# STAGE: STABLE AND GENERALIZABLE GRPO FOR AUTOREGRESSIVE IMAGE GENERATION

## ABSTRACT

Reinforcement learning has recently been explored to improve text-to-image generation, yet applying existing GRPO algorithms to autoregressive (AR) image models remains challenging. The instability of the training process easily disrupts the pretrained model capability during long runs, resulting in marginal gains, degraded image quality, and poor generalization. In this work, we revisit GRPO for AR image generation and identify two key issues: contradictory gradients from unnecessary tokens and unstable policy entropy dynamics. To address these, we introduce STAGE, a stable and generalizable framework that leverages two targeted solutions: 1) Advantage/KL reweighting. Similarity-aware reweighting to alleviate conflicting updates; and 2) Entropy reward. An entropy-based reward corresponding to reference model to stabilize learning. With the help of alleviating conflicts between tokens and an entropy reward for stabilizing training, we reduce disruption of the pretrained distribution and mitigate reward hacking, which in turn improves generalization and transfer better to other benchmarks. Experiments across multiple benchmarks show that STAGE consistently improves visual quality, stability, and cross-task generalization compared to baseline GRPO.

## 1 INTRODUCTION

Reinforcement learning (RL) for large language models (LLMs) has markedly improved performance on reasoning-intensive tasks such as mathematics and code generation. In particular, Group Relative Policy Optimization (GRPO) (Shao et al., 2024) eliminates the value model in PPO (Schulman et al., 2017), resulting in a simpler and more efficient training paradigm via group-relative advantages. Recent advances further strengthen this with importance weighting (Zheng et al., 2025; Zhao et al., 2025) and entropy regularization (Cui et al., 2025; Wang et al., 2025b), establishing RL as a powerful paradigm for performance gains and human-preference alignment.

Corresponding RL technique have also been explored for visual generation. For continuous-representation models, Liu et al. (2025b); Xue et al. (2025) investigated GRPO for flow models (Esser et al., 2024b; Labs, 2024). For discrete autoregressive (AR) approaches, T2I-R1 (Jiang et al., 2025) pairs semantic reasoning with RL, AR-GRPO (Yuan et al., 2025) directly applies GRPO for AR, Janus-focusdiff (Pan et al., 2025) integrates RL into fine-tuning, and SimpleAR (Wang et al., 2025a) applies RL in unified vision–language models to improve image quality. RL has enhanced current generative foundation models and offers a promising direction for visual reasoning.

Nevertheless, current GRPO adaptations for AR image generation largely follow LLM practices and do not fully account for the characteristics of visual tokens. In GRPO, at each RL iteration the policy generates a group of samples per prompt, a reward (e.g., HPS (Wu et al., 2023) or GenEval (Ghosh et al., 2023)) scores each sequence, and scores are propagated token-wise to update the policy. Yet, **visual tokens differ fundamentally from text**: 1) Although discrete, they represent continuous semantics align with low-level patterns in the decoded image. 2) Rollouts from the same prompt often share highly similar content, especially background regions (see Fig. 1 (b)). However, when GRPO enforces divergent updates, semantically similar regions across rollouts may be assigned opposite rewards, introducing noisy and contradictory gradients. 3) Due to the discrete and sequential nature of AR generation, AR models are highly sensitive to small distribution shifts. Especially under repeated RL optimization, the model struggles to maintain a stable distribution, leading to reward

hacking or degraded outputs and poor generalization (see Fig. 1(a)). KL regularization can partially alleviate this issue, yet some risk of performance degradation remains.

To address these challenges, we propose STAGE, which augments GRPO paradigm with two targeted improvements: 1) similarity-aware advantage/KL reweighting to improve training efficiency, and 2) an entropy-based regularizer to stabilize learning. Specifically: 1) Token-wise advantage leverages similarities among token embeddings within rollouts of the same prompt to dynamically adjust per-token advantages, reducing updates on redundant similar background tokens, mitigating conflicting gradients between positive and negative samples, and better preserving the original model capability. A similarity-aware KL schedule further suppresses unnecessary updates in irrelevant regions. 2) Entropy-based regularization calculates the entropy gap between current and reference policies and incorporates it as an auxiliary reward, further discouraging abrupt entropy drops and stabilizing policy updates. Together, these mechanisms produce a more stable and efficient RL process (see Fig. 6), mitigating reward hacking and improving generalization across image-quality metrics.

Extensive experiments on GenEval, T2I-CompBench and HPS show that our method outperforms baseline GRPO in stability and generalization. With proposed approach, Janus-Pro's GenEval score rises from 0.78 to 0.89, significantly surpassing most current diffusion and AR models. Training under HPS, OCR, and other rewards further demonstrates improvements in image quality and text rendering. Notably, our method maintains stable entropy during RL while improving GenEval performance, achieving a favorable balance between visual detail, structural consistency, and prompt adherence. Compared with the baseline, it shows stronger generalization to prompts outside the training distribution. We summarize our contributions as follows:

1. Motivated by the challenge that GRPO for autoregressive image generation often suffers from unstable training and poor generalization, we propose STAGE, which addresses contradictory gradients and unstable entropy during training, improving efficiency while mitigating instability.

2. Specifically, to handle contradictory gradients, we exploit similarities among multiple rollouts of the same prompt to avoid updates in regions shared by positive and negative samples, providing a smoother and more efficient RL process. An additional entropy-based reward that regularizes the current and reference policies further stabilizes training.

3. Experiments across diverse rewards and benchmarks show that proposed method stabilizes training and improves detail and structural consistency in generated images. It also shows better generalization than baseline to evaluation metrics and prompts out of training distribution.

## 2 RELATED WORKS

### 2.1 AUTOREGRESSIVE IMAGE GENERATION

For AR image generation, images are first quantized into discrete tokens (Esser et al., 2021; Yu et al., 2021) and then generated with Transformers in raster order (Ding et al., 2021; Ge et al., 2023; Ramesh et al., 2021; Yu et al., 2022; He et al., 2024; Wang et al., 2024). Recent efforts have scaled this paradigm with larger models and stronger conditioning. LlamaGen (Sun et al., 2024) provides class and text-conditioned baselines; while Liu et al. (2024a) and Chern et al. (2024) fine-tune Chameleon (Team, 2025) for improved text-conditioned generation.

Recent work has explored unified vision-language generation, producing images and text within a single transformer (Wu et al., 2024; Chen et al., 2025; Jiao et al., 2025; Ma et al., 2025; Zhang et al., 2025; Qu et al., 2025), more powerful tokenizers (Lee et al., 2022; Yu et al., 2023), and strategies for parallel multi-token generation or token compression (Tian et al., 2024; Ma et al., 2024; Yu et al., 2024; Liu et al., 2025c; 2024b; 2025a). Training AR models typically involves multiple stages to better utilize limited high-quality data, and prior studies (Sun et al., 2024; Chen et al., 2025) emphasize that carefully designed curricula are crucial for strong generative performance.

### 2.2 REINFORCEMENT LEARNING

Reinforcement learning (RL) has been widely adopted in large language models (LLMs) to improve reasoning and alignment (DeepSeek-AI et al., 2025). GRPO (Shao et al., 2024) simplifies policy optimization by removing the value model in PPO (Schulman et al., 2017) and achieves strong

empirical gains using a relative group-based objective. Subsequent refinements, including importance sampling (Zheng et al., 2025; Zhao et al., 2025), gradient clipping (Yu et al., 2025) and entropy-based regularization (Cui et al., 2025; Wang et al., 2025b), which further stabilize training.

In visual generation, RL has been used to enhance fidelity and controllability. Flow-based approaches (Liu et al., 2025b; Xue et al., 2025) apply GRPO to align continuous generative processes (Labs, 2024; Esser et al., 2024b) with human preferences (Kirstain et al., 2023; Wu et al., 2023) or prompt alignment (Huang et al., 2023; Ghosh et al., 2023). For AR models, RL is applied differently: T2I-R1 (Jiang et al., 2025) uses semantic reasoning to improve text-to-image alignment, AR-GRPO (Yuan et al., 2025) provides a direct AR+GRPO baseline, and SimpleAR (Wang et al., 2025a) integrates RL into unified model to improve quality. Despite these advances, current GRPO for AR image generation still suffers from inefficiency, unstable training and sub-optimal generalization.

## 3 METHOD

### 3.1 PRELIMINARIES

**Autoregressive image generation.** In a standard autoregressive (AR) generation pipeline, an image $I \in \mathbb{R}^{H \times W \times 3}$ is discretized into a sequence of tokens $(x_1, x_2, \ldots, x_{h \times w})$, where each token $x_t \in [V]$ corresponds to an index in a learned codebook of size $V$ (e.g., from a VQ-VAE tokenizer). Given corresponding text tokens $c$, the transformer is trained to model the joint distribution of image tokens in a flattened sequence, where causal attention restricts each position to attend only to preceding tokens. During generation, tokens are produced in a raster-scan manner (from top-left to bottom-right). At step $t$, the model predicts a categorical distribution over the vocabulary conditioned on all previously generated tokens $x_{1:t-1}$. The overall generation process can thus be factorized as:

$$p(x_{1:h \times w}) = \prod_{t=1}^{h \times w} p(x_t \mid x_{1:t-1}; c), \tag{1}$$

where $p(x_t \mid x_{1:t-1}; c)$ represents the conditional distribution of the $t$-th token on previous $t-1$ image and text tokens. This sequential factorization models long-range dependencies across visual tokens but also makes generation sensitive to distributional shifts accumulated along the sequence.

**Group relative policy optimization (GRPO).** For image generation, GRPO improves downstream metrics via an iterative *generate–evaluate–update* process. Concretely, the policy model $\pi_\theta$, parameterized as an AR transformer, produces $G$ diverse image outputs $\{o_1, \ldots, o_G\}$ conditioned on $c$. Each output is then scored by a reward function $\mathcal{R}(\mathbf{x}, c)$ to obtain rewards $\{R_1, \ldots, R_G\}$. Advantages for each token $t$ in sample $i$ are computed by normalizing rewards within the group:

$$\hat{A}_{i,t} = \frac{R_i - \text{mean}(\{R_j\}_{j=1}^G)}{\text{std}(\{R_j\}_{j=1}^G)}. \tag{2}$$

Subsequently, using importance sampling, policy $\pi_\theta$ is updated by maximizing following objective:

$$\mathcal{J}_{\text{GRPO}}(\theta) = \mathbb{E}\left[ \frac{1}{G} \sum_{i=1}^G \frac{1}{|o_i|} \sum_{t=1}^{|o_i|} \min\left(r_{i,t}(\theta)\hat{A}_{i,t}, \, \text{clip}(r_{i,t}(\theta), 1-\varepsilon, 1+\varepsilon)\hat{A}_{i,t}\right) - \beta D_{\text{KL}}(\pi_\theta \| \pi_{\text{ref}}) \right], \tag{3}$$

where $\pi_{\text{ref}}$ denotes the reference policy for regularization and prevent from distributional drift. The importance ratio $r_{i,t}(\theta)$ is defined as:

$$r_{i,t}(\theta) = \frac{\pi_\theta(o_{i,t}|o_{i,<t}; c)}{\pi_{\theta_{\text{old}}}(o_{i,t}|o_{i,<t}; c)}, \tag{4}$$

which measures the relative likelihood of token $o_{i,t}$ under current policy $\pi_\theta$ and old policy $\pi_{\theta_{\text{old}}}$. For brevity, we omit the mapping from token sequences to rendered images. Here, $o_{i,t}$ denotes the $t$-th token of $i$-th sample, and $o_i$ is $i$-th generated image or corresponding full token sequence.

### 3.2 ADVANTAGE & KL REWEIGHT

**Instability during AR training.** During GRPO training of AR image generation models, we observe persistent instability. For example, under the GenEval reward, late-stage RL often produces

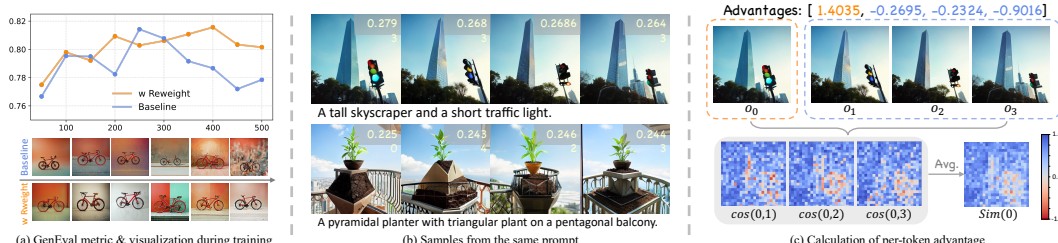

(a) GenEval metric & visualization during training  (b) Samples from the same prompt  (c) Calculation of per-token advantage

Figure 1: (a) Distributional disruption caused by conflicting gradients during training, especially with large learning rates and weak KL loss (lr = 5e-6, GenEval reward, KL weight = 0.01). (b) Multiple samples generated from the same prompt exhibit high similarity, illustrated with Janus-pro 7B. (c) Pairwise cosine similarity of VQ embeddings across images generated with the same prompt.

structurally degraded images and plateaued evaluation scores, particularly with large learning rates, many iterations, or low KL loss weight. We attribute this to disruption of the pretrained model distribution (see Fig. 1(a)). As training progresses, the model's prior knowledge of specific concepts, such as bicycles, is gradually degraded, resulting in deteriorated generated structures.

Addressing this issue requires better preservation of the pretrained distribution during RL training. We attribute this to unstable training and noisy or conflicting gradients in RL objective, which obscure the optimization direction and cause distributional drift, ultimately degrading training outcomes.

**The conflicts in generated images.** As discussed in Sec. 3.1, at each training step the policy $\pi_\theta$ generates multiple outputs $\{o_i\}$ conditioned on text $c$. For AR visual generation, outputs from the same prompt often share highly similar regions, differing only in fine details (see Fig. 1 (b)).

Unlike text tokens in LLMs, visual tokens exhibit strong local similarity. Although different images may receive different advantage values, many regions across outputs are nearly identical (even if token indices differ, their VQ embeddings are close), which results in conflicting gradients when advantages have opposite signs.

To quantify this, we compute token-level cosine similarity between VQ embeddings. For images $o_i$ and $o_j$, with embeddings $q_{i,t}, q_{j,t} \in \mathbb{R}^C$ at position $t \in \{1, \ldots, h \times w\}$, we define and rewrite $\cos(q_{i,t}, q_{j,t})$ as $\cos(i, j, t)$, the cosine similarity is calculated as:

$$\cos(i, j, t) = \frac{q_{i,t} \cdot q_{j,t}}{\|q_{i,t}\| \, \|q_{j,t}\|}. \tag{5}$$

As shown in Fig. 1 (c), regions with similar content exhibit notably higher similarity scores. Even when cosine similarity is only slightly above zero, corresponding regions remain visually alike, while in most areas both cosine similarity and visual content across images are consistently high. Motivated by this observation, we aim to discard redundant tokens or reduce their update during training.

**Differences between text and image tokens.** Due to the nature of next-token prediction and its reliance on sampling from the logits distribution, AR generation is highly sensitive to distribution shifts: injecting mild Gumbel noise into the logits can substantially alter the generated image structure (Fig. 2 (a)), showing that AR decoding over discrete image codes is similarly distribution sensitive.

In contrast, because images are continuous rather than truly discrete, VQ decoding is highly robust to token indices: many indices encode similar low-level features, and replacing tokens with their top-$k$ nearest neighbors still yields nearly identical images (Fig. 2 (b)). This mismatch may lead to current GRPO that optimizes on specific index to assign opposite gradients to semantically similar regions with different indices, introducing noise and limiting training efficiency.

***Solution*: Advantage & KL reweighting.** Based on the previous observation, we leverage embedding similarity between samples with opposite advantage signs to identify and mitigate conflicting gradients. For a group of images $\{o_1, \ldots, o_G\}$ normalized advantages $\{\hat{A}_{i,t}\}_{i=1}^{G}$, For each sample $i$ at position $t$, we compute the cosine similarity $\cos(i,j,t)$ based on Eq. 5, and aggregate similarities only with tokens from samples $j$ whose advantages have opposite signs:

$$\text{Sim}(i, t) = \frac{\sum_{j=1}^{G} \mathbf{1}[\hat{A}_{i,t}\hat{A}_{j,t} \leq 0] \cdot \cos(i, j, t)}{\sum_{j=1}^{G} \mathbf{1}[\hat{A}_{i,t}\hat{A}_{j,t} \leq 0]}. \tag{6}$$

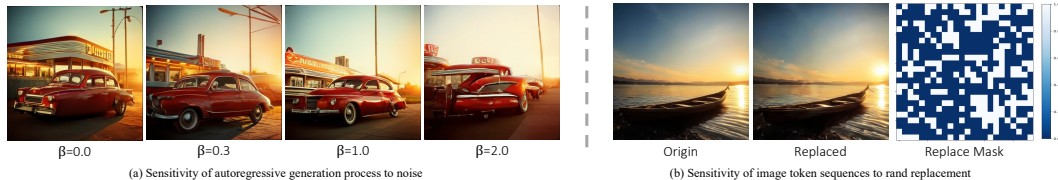

| β=0.0 | β=0.3 | β=1.0 | β=2.0 | | Origin | Replaced | Replace Mask |

(a) Sensitivity of autoregressive generation process to noise ⋮ (b) Sensitivity of image token sequences to rand replacement

Figure 2: (a) AR generation is highly sensitive to distribution perturbations—injecting Gumbel noise $\beta g$, $g \sim \text{Gumbel}(0, 1)$, easily disrupts the decoded image semantics. In contrast, VQ tokenization is robost to token index: replacing $>50\%$ tokens in image sequence with top-50 nearest token still yields an almost visually identical image.

We then define a soft mask to down-weight highly similar tokens:

$$M_{i,t} = \text{Norm}(1 - \text{Sim}(i, t)), \tag{7}$$

where $\text{Norm}(\cdot)$ scales and shifts $M_{i,t}$ from range $[-1, 1]$ to $[0, 1]$. And the masked advantage is then

$$\tilde{A}_{i,t} = M_{i,t} \cdot \hat{A}_{i,t}. \tag{8}$$

By replacing $\hat{A}_{i,t}$ in Eq. 2 with the weighted form $\tilde{A}_{i,t}$, we obtain the similarity-modulated advantage. Intuitively, tokens similar to opposite-advantage tokens are down-weighted to avoid contradict gradients, while less similar tokens are amplified for clearer optimization direction.

To further stabilize training, we dynamically reweight the per-token KL penalty using embedding similarity. For each position, the KL weight is scaled according to the embedding distance between the positive and negative samples: when the embeddings are close (high model certainty), we apply a larger KL penalty, and vice versa. Based on the similarity $\text{Sim}(i, t)$ from Eq. 6, the per-token KL weight is computed as $\beta'_{i,t} = (a + b \cdot \text{clip}(\text{Sim}(i, t) + 1, 0, 1)) \beta$, where $a = b = 0.5$.

### 3.3 ENTROPY REWARD FOR STABLIZED TRAINING

**Entropy & generated images.** We first define entropy in the context of autoregressive generation. Given a policy $\pi_\theta$ and conditional distribution over token vocabulary at step $t$, entropy is defined as:

$$\mathcal{H}_t = -\sum_{x \in V} \pi_\theta(x \mid x_{<t}; c) \log \pi_\theta(x \mid x_{<t}; c), \tag{9}$$

where $V$ denotes the vocabulary and $c$ the input condition. The overall entropy of a generated sequence $o = (o_1, \ldots, o_T)$ is obtained by averaging across positions:

$$\mathcal{H}(o) = \frac{1}{T} \sum_{t=1}^{T} \mathcal{H}_t. \tag{10}$$

Entropy plays a crucial role in AR image generation. A low-entropy policy tends to produce highly deterministic outputs, possibly leading to a loss of diversity, while an excessively high-entropy policy encourages randomness that results in noisy or semantically inconsistent generations.

To study this relationship, we perform a controlled study by varying sampling temperature $\tau$ of a fixed AR model during generation, which directly controls token-level entropy. As shown in Fig. 3, decreasing $\tau$ makes the model's sampling overly confident, which reduces content richness and often degrades image quality, whereas increasing $\tau$ encourages more diverse content at the cost of structural fidelity. (See Appendix D.1 for additional analysis). These results indicate that maintaining an appropriate entropy range is essential for balancing fidelity and diversity in generation, and motivate the introduction of entropy-aware reward during RL fine-tuning.

**Entropy collapse during RL.** During RL training, the policy entropy of each generated sample $\{o_1, \ldots, o_G\}$ is defined based on the probability distributions of all tokens obtained by feeding the generated sample into the policy $\pi_\theta$, and computed according to Eq. 9 and Eq. 10.

Training with different rewards induce distinct policy entropy dynamics. For VQA- or rule-based rewards (e.g., GenEval in Flow-GRPO (Liu et al., 2025b)), qualitatively different images can receive

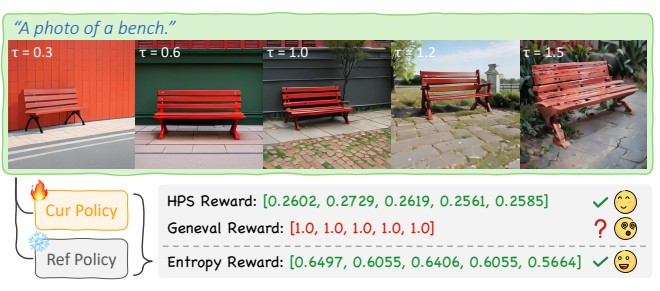

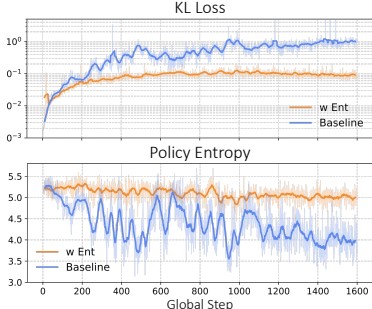

Figure 3: Samples from same prompt generated by varying temperature $\tau$. HPS gives discriminative scores for high-quality images; rule-based GenEval returns identical scores, causing confusion and instability. Entropy reward favors samples with entropy closer to reference model, giving a clear preference.

Figure 4: With GenEval reward, unstable policy entropy leads to rapid KL growth and unstable training ("Baseline"), entropy reward stabilizes entropy and KL ("w Ent").

the same reward—as long as the required object appears, the output is marked correct (Fig. 3). This creates ambiguity for the policy, leading to prompt-dependent entropy behaviors and frequent entropy fluctuations during training (Fig. 4). Such instability reduces image diversity and visual quality, eventually causing model degradation. Similar entropy-collapse phenomena have been observed in text-based GRPO (Cui et al., 2025; Wang et al., 2025b).

***Solution*: Entropy reward for stabilized training.** As discussed above, the reward model often produces hard signals without smooth variations among samples, causing confusion and unstable policy entropy. To mitigate this effect, we introduce a reward item based on policy entropy for the samples with the highest reward. Specifically, given the predicted token of sample $o_i$ distributions from current policy $\pi_\theta$ and reference policy $\pi_{\text{ref}}$, we compute their per-token entropy according to Eq. 10, yielding $\mathcal{H}_\theta(o_i)$ and $\mathcal{H}_{\text{ref}}(o_i)$. The entropy reward is then defined as:

$$R_i^{\text{ent}} = \left(1 + \left(\Delta\mathcal{H}_i\right)^2\right)^{-1}, \quad \text{where } \Delta\mathcal{H}_i = \mathcal{H}_{\text{ref}}(o_i) - \mathcal{H}_\theta(o_i); \tag{11}$$

and is added to the original reward. Note that, to avoid potential influence on the final results, we apply the entropy reward only to the top-rewarded samples:

$$R_i' = R_i + \lambda \cdot R_i^{\text{ent}} \cdot \mathbf{1}[R_i = \max_j R_j], \tag{12}$$

where $\lambda$ is a weighting coefficient and $\mathbf{1}[\cdot]$ is the indicator function. We set $\lambda=0.4$ in our experiments. This entropy reward encourages the policy to maintain a level of uncertainty comparable to the reference model and prevents entropy collapse. As a result, it mitigates potential instability and reduces distributional drift relative to the reference policy (see Fig. 4). Training with the entropy reward yields markedly more stable entropy trajectories and substantially lower KL loss. Additional analysis of entropy reward can be found in Appendix D.2.

## 4 EXPERIMENT

### 4.1 IMPLEMENTATION DETAILS

We build our experiments on the Janus-Pro 7B model (Chen et al., 2025) and evaluate several types of rewards: (1) GenEval-rules rewards, which measure how much RL improves the model's prompt-following ability; (2) A mixture of human-preference, object-detector, and VQA rewards (HPS (Wu et al., 2023) + Gdino (Liu et al., 2024c) + Git (Wang et al., 2022)), following the T2I-R1 setup (Jiang et al., 2025) to assess human preference and aesthetics; and (3) An OCR-based reward (Gong et al., 2025), computed as the minimum edit distance between the generated text and the target text, to evaluate text-rendering capability.

Different types of rewards use different prompt formats during training. For GenEval reward, we follow the same strategy as Flow-GRPO (Liu et al., 2025b) and adopt prompt format used in GenEval benchmark. For the mixed reward, we continue to use prompting strategy in (Jiang et al., 2025). For OCR reward, we use prompts that include quotation marks to explicitly specify the text to be rendered. Additional experimental details and parameter settings are provided in Appendix B.1.

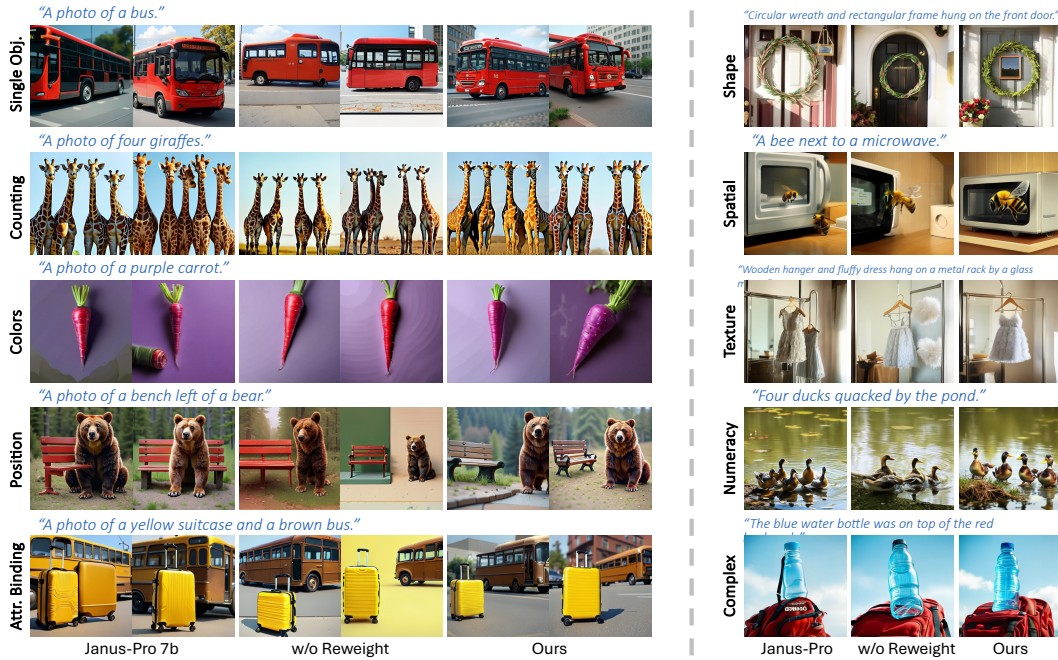

Figure 5: Generations comparison on GenEval (left) and T2I-CompBench (right) prompts. Compared to Baseline GRPO (with entropy reward to stablize training, labeled as "w/o Reweight"), our advantage/KL reweighting preserves the base distribution and improves concept accuracy, yielding stronger structural stability, layout consistency, and finer detail than GRPO baselines.

Table 1: Quantitative results on GenEval across models. Compared to vanilla GRPO ("Baseline"), adding only entropy reward ("+Ent", i.e., Baseline + Entropy) noticeably stabilizes training. Our full method ("Ours") further adds advantage/KL reweighting based on "+Ent", brings additional improvements; we provide GenEval of our last checkpoint ("Ours") and best ("Ours*") for reference.

| Model | Overall↑ | Single Obj.↑ | Two Obj.↑ | Counting↑ | Colors↑ | Position↑ | Attr. Binding↑ |
|---|---|---|---|---|---|---|---|
| Pixart-$\alpha$ (2023) | 0.48 | 0.98 | 0.50 | 0.44 | 0.80 | 0.08 | 0.07 |
| SD3 (2024a) | 0.74 | 0.99 | 0.94 | 0.72 | 0.89 | 0.33 | 0.60 |
| FLUX.1-dev (2024) | 0.66 | 0.98 | 0.79 | 0.73 | 0.77 | 0.22 | 0.45 |
| Sana-1.5 (2025) | 0.81 | 0.99 | 0.93 | 0.86 | 0.84 | 0.59 | 0.65 |
| LlamaGen (2024) | 0.32 | 0.71 | 0.34 | 0.21 | 0.58 | 0.07 | 0.04 |
| Show-o (2024) | 0.68 | 0.98 | 0.80 | 0.66 | 0.84 | 0.31 | 0.50 |
| Infinity (2024) | 0.73 | - | 0.85 | - | - | 0.49 | 0.57 |
| GPT-4o (2024) | 0.85 | 0.99 | 0.92 | 0.85 | 0.91 | 0.75 | 0.66 |
| Janus-Pro-7B (2025) | 0.78 | 0.98 | 0.86 | 0.56 | 0.89 | 0.76 | 0.63 |
| T2I-R1 (2025) | 0.79 | 0.99 | 0.91 | 0.53 | 0.91 | 0.76 | 0.65 |
| Baseline | 0.86 | 0.97 | 0.92 | 0.82 | 0.86 | 0.84 | 0.72 |
| +Ent | 0.87 | 0.99 | 0.94 | 0.78 | 0.90 | 0.89 | 0.73 |
| Ours | 0.88 | 0.99 | 0.93 | 0.82 | 0.89 | 0.91 | 0.77 |
| Ours* | 0.89 | 0.99 | 0.95 | 0.82 | 0.90 | 0.89 | 0.79 |

## 4.2 MAIN RESULTS AMONG METRICS

We evaluate ours against baselines on metrics (GenEval, T2I-Compbench, etc.). Note that "Baseline" refers to vanilla GRPO, while "Ours" denotes the method with proposed dynamic advantage/KL reweighting and entropy reward (the latter applied only for GenEval reward).

**GenEval.** For GenEval reward, we provide both the last and best result and overall performance curves with respect to iteration (Table 1 and Fig. 5). The baseline GRPO improves the GenEval score from 0.78 to 0.86, while our method further boosts it to 0.89, surpassing many existing diffusion and AR models. In comparison, the T2I-R1 scheme brings only about 0.01 improvement on Janus-Pro.

**T2I-Compbench.** For model trained with HPS+Gdino+Git mixed reward, we report T2I-Compbench in Table 2 ("Baseline" and "Ours"), which demonstrates the stability improvement of structures and layouts in generated images. Additionally, we report the generalization of models trained with the GenEval reward on T2I-Compbench (Fig. 5); entries marked with "†" in Table 2 demonstrate that our

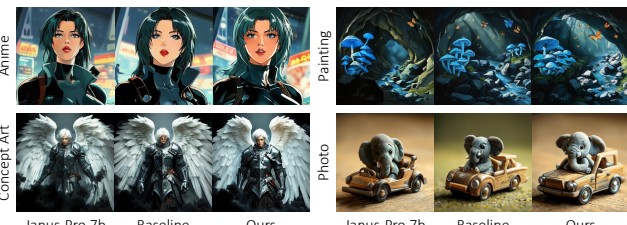

Figure 6: GenEval vs. global steps during RL. Ours converges faster and attains higher final performance than baseline GRPO with entropy reward.

Figure 7: Visualize comparison on HPS prompts. Our method delivers clearer fine details and stronger structural stability than both the baseline GRPO and the original Janus-Pro 7b (see human face, armors and structure of cars).

Table 2: Quantitative results on T2I-compbench. Compared to vanilla GRPO ("Baseline"), our models trained with the GenEval reward generalize better, yielding consistent gains across multiple sub-benchmarks. "Spat." is short for spatial. "†" means model is trained on GenEval reward.

| Model | Color↑ | Shape↑ | Texture↑ | 2D-Spat.↑ | 3D-Spat.↑ | Non-Spat.↑ | Numeracy↑ | Complex↑ |
|---|---|---|---|---|---|---|---|---|
| SD3 (2024a) | 0.8094 | 0.5864 | 0.7297 | 0.3219 | 0.4044 | 0.3143 | 0.6078 | 0.3780 |
| FLUX.1-dev (2024) | 0.7407 | 0.5718 | 0.6922 | 0.2863 | 0.3866 | 0.3127 | 0.6185 | - |
| Sana-1.5 (2025) | 0.7625 | 0.5426 | 0.6761 | 0.3814 | 0.4088 | 0.3123 | 0.6110 | 0.3727 |
| LlamaGen (2024) | 0.2996 | 0.3212 | 0.3888 | 0.1004 | 0.1530 | 0.2729 | 0.2747 | 0.2501 |
| Show-o (2024) | 0.7327 | 0.5264 | 0.6815 | 0.3697 | 0.3996 | 0.3106 | 0.6209 | 0.3572 |
| Infinity (2024) | 0.7379 | 0.4650 | 0.5919 | 0.2215 | 0.3846 | 0.3076 | 0.5475 | 0.3689 |
| Janus-Pro-7B (2025) | 0.6355 | 0.3494 | 0.4929 | 0.1931 | 0.3279 | 0.3087 | 0.4423 | 0.3566 |
| T2I-R1 (2025) | 0.8130 | 0.5852 | 0.7243 | 0.3378 | - | 0.3090 | - | 0.3993 |
| Baseline† | 0.7143 | 0.4028 | 0.6085 | 0.2763 | 0.3692 | 0.3090 | 0.4394 | 0.3586 |
| Ours† | 0.7463 | 0.4388 | 0.6443 | 0.3053 | 0.3667 | 0.3107 | 0.5278 | 0.3779 |
| Baseline | 0.7829 | 0.5842 | 0.7380 | 0.3674 | 0.4042 | 0.3131 | 0.5902 | 0.4004 |
| Ours | 0.7842 | 0.5923 | 0.7451 | 0.3731 | 0.4005 | 0.3136 | 0.5993 | 0.3997 |

method improves transfer performance. Notably, for the Numeracy evaluation dimension, the score rises from 0.43 to 0.52, even though the prompts are not aligned with those used in T2I-Compbench. Additional visual comparison can be found in Appendix Fig. 26.

**HPS & ImageReward.** We report HPS and ImageReward scores for models trained with the mixed HPS+Gdino+Git reward. Compared to the non-reweighted baseline, our method ("Ours") achieves higher scores and noticeably better visual quality (Table 3, Fig. 7). While vanilla GRPO improves fine-grained details, dynamic weighting further enhances structural accuracy and rendering fidelity. Additional metrics (aesthetic, pickscore, DeQA) are provided in Appendix Table 8.

**OCR.** For OCR reward, we follow Flow-GRPO and evaluate on its OCR test set (∼1,000 text-generation prompts) using average text edit distance (see Fig. 8 and Table 3). Ours yields more stable text generation, whereas baseline GRPO sometimes underperforms the original Janus-Pro in visual examples. The RL-trained model surpasses discrete AR and many diffusion models.

**Generalization evaluation.** We attribute the improved generalization to a better balance between the base model's original distribution and the RL updates, which prevents distribution drift and leads to

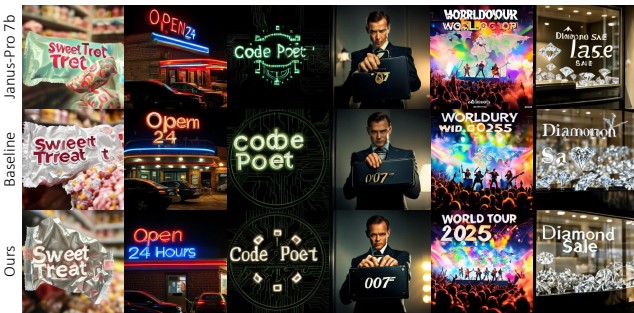

Figure 8: Visualization of text rendering capability. Compared with the baseline, ours more accurately captures the textual structure while maintaining image generation quality.

Table 3: Quantitative evaluation of baseline GRPO on and ours on HPS, ImageReward, and text rendering.

| Model | HPS↑ | ImgRwd↑ | OCR↑ |
|---|---|---|---|
| Pixart-α (2023) | 30.76 | 0.75 | 0.04 |
| SD3 (2024a) | 30.22 | 1.00 | 0.57 |
| FLUX.1-dev (2024) | 31.35 | 1.10 | 0.63 |
| Sana-1.5 (2025) | 30.36 | 1.08 | 0.33 |
| LlamaGen (2024) | 23.92 | -0.36 | 0.04 |
| Show-o (2024) | 27.98 | 0.86 | 0.08 |
| Infinity (2024) | 30.60 | 0.88 | 0.36 |
| Janus-Pro-7B (2025) | 28.64 | 0.76 | 0.21 |
| T2I-R1 (2025) | 29.83 | 0.94 | 0.23 |
| Baseline | 29.73 | 0.93 | 0.46 |
| Ours | 29.87 | 0.98 | 0.49 |

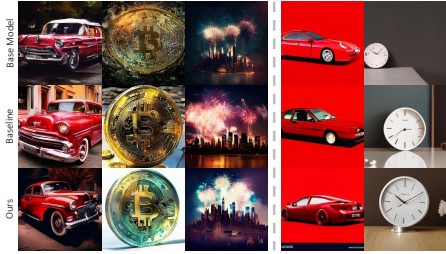

Figure 9: Visualization of methods applied to STAR (left) and LlamaGen (right). Ours enhances structural stability and better preserves fine-grained details than baseline.

Table 4: Effect of our method on other base models. We evaluate on T2I-Compbench, HPS, ImageReward, and GenEval, where our approach improves generation quality more effectively than the Baseline, achieving consistent gains across various metrics.

| Model | T2I-Compbench↑ | | | HPS↑ | ImgRwd↑ | GenEval↑ |
|---|---|---|---|---|---|---|
| | Text. | 2D-Spat. | Num. | | | |
| LlamaGen | 0.5041 | 0.0813 | 0.3761 | 21.27 | -0.31 | 0.32 |
| Baseline | 0.5326 | 0.0769 | 0.3901 | 22.53 | -0.22 | 0.35 |
| Ours | 0.5554 | 0.0961 | 0.4109 | 22.74 | -0.17 | 0.39 |
| STAR | 0.5393 | 0.1627 | 0.4987 | 26.38 | 0.51 | 0.47 |
| Baseline | 0.5849 | 0.1967 | 0.5249 | 30.17 | 0.76 | 0.49 |
| Ours | 0.6013 | 0.2063 | 0.5456 | 30.50 | 0.89 | 0.51 |

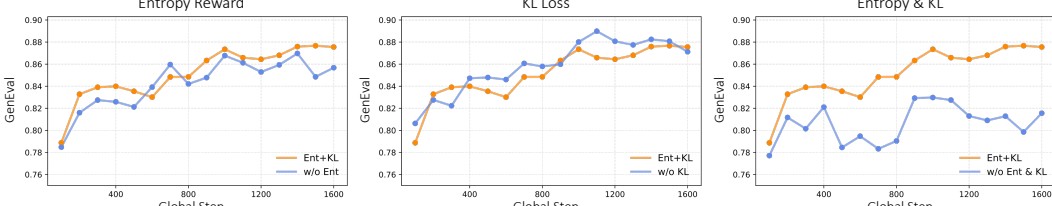

Figure 10: Ablations on Geneval: from a Baseline+Entropy reward setting ("Ent+KL"), dropping the KL on zero-variance groups ("w/o KL") or the entropy reward ("w/o Ent") reduces training stability.

higher image quality. Beyond T2I-Compbench (Table 2), we also evaluate GenEval-trained models on image-quality metrics such as ImageReward. Ablations (Table 5) show that the gains come from our entropy reward and dynamic reweighting, which more effectively balance performance and generality (see Sec. 4.3). Additional generalization results—e.g., training with GenEval but evaluating under human-preference metrics from T2I-R1 (Jiang et al., 2025)—are provided in Appendix C.1.

**Experiments on other base models.** Furthermore, we validate our method on additional base models. We select LlamaGen (Sun et al., 2024), which is also based on next-token prediction, and a next-scale paradigm STAR (Ma et al., 2024). We use the second reward defined in the Implementation Details—the mixture of human-preference, object-detector, and VQA rewards. More details are provided in the supplementary material (Sec. B.1). We report the results of Baseline GRPO and Ours on T2I-Compbench, HPS, and ImageReward in Table 4 and Fig. 9. Our method consistently outperforms Baseline GRPO in prompt following, aesthetics, and related metrics.

### 4.3 ABLATIONS & DISCUSSIONS

**Impact of advantage & KL reweighting.** Fig. 6 shows that our reweighting strategy ("Ours") accelerates convergence and keeps GenEval stably above 0.88, while the version without reweighting ("w/o Reweight") degrades in later stages due to distribution drift. The reweighting stabilizes optimization, prevents collapse, leads to better generalization and finer visual details (Table 5, Fig. 5). A key effect is that reweighting reduces gradient variance among samples in the same group, preventing conflicting updates in GRPO and mitigating distribution drift (see Fig. 13 (a)).

To disentangle the contributions of each component, we further remove advantage or KL reweighting individually (see "w/o adv./KL Reweight" in Table 5). Both variants exhibit performance drops in GenEval, indicating that the two mechanisms play analogous roles—modulating update magnitude based on sample similarity: advantage reweighting scales gradients directly, whereas KL reweighting adjusts the regularization strength that constrains deviation from the reference policy.

**Impact of entropy reward.** Fig. 4 shows that entropy reward stabilizes KL loss and policy entropy, helping maintain the model's distribution and improving GenEval performance. Removing it ("w/o Ent") introduces larger fluctuations and degrades both stability and final accuracy (Fig. 10, Fig. 12).

To better understand the effect of entropy reward, we replace it with HPS under the same sample-selection protocol. As shown in Fig. 13(b) and Fig. 23, HPS drives the policy entropy to collapse early: GenEval rises briefly but soon degrades, indicating distribution drift and reward hacking under KL constraints. In contrast, the entropy reward acts as a soft, sample-level KL regularizer, keeping the

Table 5: Ablations across variants on GenEval (in-distribution), T2I-CompBench and ImageReward (out-of-distribution). Ours achieves the best in- and out-of-distribution performance over baselines.

| Model | GenEval | | | | | | | T2I-Compbench | | | | | ImgRwd |
|---|---|---|---|---|---|---|---|---|---|---|---|---|---|
| | Overall | Single | Two | Count | Color | Pos. | Attr | Color | Shape | Text. | 2d Spat. | Num. | |
| Janus-Pro 7B | 0.78 | 0.98 | 0.86 | 0.56 | 0.89 | 0.76 | 0.63 | 0.6355 | 0.3494 | 0.4929 | 0.1931 | 0.4423 | 0.76 |
| Ours | 0.88 | 0.99 | 0.93 | 0.82 | 0.89 | 0.91 | 0.77 | 0.7463 | 0.4388 | 0.6443 | 0.3053 | 0.5278 | 0.80 |
| w/o adv. Reweight | 0.87 | 0.99 | 0.94 | 0.79 | 0.88 | 0.85 | 0.75 | 0.7460 | 0.4544 | 0.6507 | 0.3120 | 0.5294 | 0.82 |
| w/o KL Reweight | 0.86 | 0.98 | 0.93 | 0.81 | 0.90 | 0.82 | 0.73 | 0.7349 | 0.4107 | 0.6226 | 0.2965 | 0.5058 | 0.76 |
| *Without reweighting (no Adv / KL reweight)* | | | | | | | | | | | | | |
| Ent+KL (w/o Reweight) | 0.87 | 0.99 | 0.94 | 0.78 | 0.90 | 0.89 | 0.73 | 0.7241 | 0.4063 | 0.6032 | 0.2788 | 0.4771 | 0.74 |
| w/o Ent (Baseline) | 0.86 | 0.97 | 0.92 | 0.82 | 0.86 | 0.84 | 0.72 | 0.7143 | 0.4028 | 0.6085 | 0.2763 | 0.4394 | 0.67 |
| w/o KL | 0.87 | 0.98 | 0.93 | 0.82 | 0.86 | 0.86 | 0.74 | 0.7214 | 0.4149 | 0.6214 | 0.3005 | 0.4923 | 0.77 |
| w/o Ent & KL | 0.81 | 0.98 | 0.90 | 0.73 | 0.88 | 0.71 | 0.67 | 0.7269 | 0.4250 | 0.6558 | 0.3072 | 0.4591 | 0.76 |

policy closer to the reference distribution and enabling continued improvement over longer training. Further analysis is provided in Appendix D.2.

**Discussion of KL loss.** KL loss is crucial for maintaining distribution of AR models, removing it may disrupt the distribution at early stages and cause reward hacking. Following DAPO (Yu et al., 2025), we skip KL terms when group reward variance is zero. This stabilizes reward usage but introduces larger training fluctuations ("w/o KL" in Fig. 10) and may degrade generation quality (Fig. 12).

**Diversity of generated content.** The proposed sample-similarity weighting may raise concerns regarding a potential reduction in diversity of generated images. However, results on GenEval reward indicate that, by better preserving original distribution, the weighting can in fact enhance diversity in certain cases. Nevertheless, instances of decreased diversity do exist (see Fig. 11). A more detailed discussion on diversity is provided in the Appendix D.3.

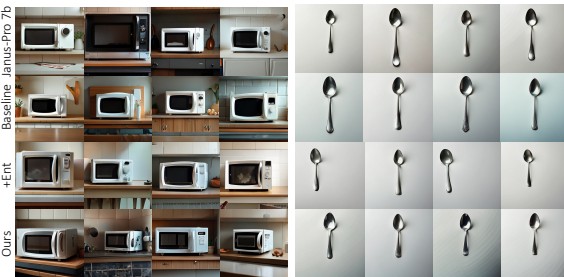

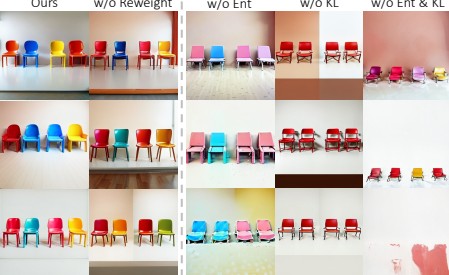

Figure 11: Visualization of diversity. Baseline GRPO reduce diversity due to distribution drift, entropy reward ("+Ent") mitigates it. "Ours" may drop diversity under strong foreground–background contrast.

Figure 12: Removing entropy reward ("Baseline") or zeroing KL for equal-reward groups worsens images compared to "Ours" and "w/o Reweight".

# 5 CONCLUSION

In this work, we investigate limitations of existing GRPO for AR text-to-image method and propose dynamic weighting strategy based on characteristics of image tokens and AR image generation to avoid redundant gradients that perturb the model's learned distribution. We further introduce an entropy-based reward to stabilize training. Together, these techniques improve training stability and gener-

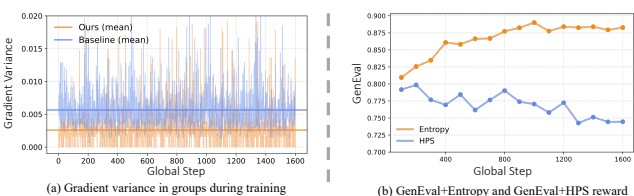

(a) Gradient variance in groups during training

(b) GenEval+Entropy and GenEval+HPS reward

Figure 13: (a) Advantage/KL reweighting suppresses intra-group gradient variance, avoiding conflicting gradients and stabilizing optimization. (b) Using HPS ("HPS") to replace entropy reward ("Entropy") causes distribution drift and hinders GenEval improvement over training.

alizability, raising the performance ceiling of AR generative models. We hope this work inspires further research toward closing the gap between AR and diffusion-based methods.

# 6 ETHICS STATEMENT

Our work studies autoregressive image generation and reinforcement learning with automated reward signals. We use only publicly available datasets and evaluation suites that are commonly used in the community and that, to our knowledge, do not contain personally identifiable information. No human subjects research was conducted. Licenses for all third party datasets, models, and evaluation tools are respected and cited in the paper and appendix.

Generative models can reproduce social biases or generate inappropriate content if prompted adversarially. Automated reward models such as HPS, ImageReward, Grounding DINO, GIT, and GenEval may themselves encode biases. We report failure cases and recommend deploying our method together with safety filtering, content moderation, and prompt auditing. Our method is not intended for the creation of deceptive, harmful, or illegal content.

We are mindful of environmental impacts. We reuse publicly released checkpoints where possible, limit ablation sweeps, and will report hardware, runtime, and estimated energy use to enable fair comparison and reduce redundant computation.

# 7 REPRODUCIBILITY STATEMENT

We aim for full reproducibility. Here we provide contents for better reproducibility:

1. Hyperparameters (e.g., the coefficient of the entropy reward) are described in Sec. 3.2 Sec. 3.3.
2. A brief overview of the base model, training and optimizer settings, and reward functions is provided in Sec. 4.1, with full training details in Appendix B.1.

We also commit to releasing all model training resources opensource soon, including the datasets used, training configurations, our pretrained weights and the corresponding codebase to reproduce our experimental results.

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

## A  ADDITIONAL METHOD DESCRIPTION

In one GRPO iteration, the current policy first generates a group of samples $\{o_1, o_2, \ldots\}$. The generated samples are (i) scored by the reward rule to obtain sequence-level rewards $\{R_1, R_2, \ldots\}$ and (ii) fed to both the current policy and the reference policy to obtain the corresponding log-probabilities and the two policy entropies $H_\theta$ and $H_{\text{ref}}$. We compute the entropy reward according to Eq. 11 and the combined reward $R'$ according to Eq. 12. After normalization we obtain the advantages $\hat{A}$, which are reweighted into $\tilde{A}$ and KL coefficients following Eqs. 6∼8. Using the log-probabilities and the KL schedule we compute the importance ratios $r_{i,t}(\theta)$ as in Eq. 4, and finally form the GRPO objective $J_{\text{GRPO}}(\theta)$ for the current update using Eq. 3. A brief visualization of our framework can be found in Fig. 14.

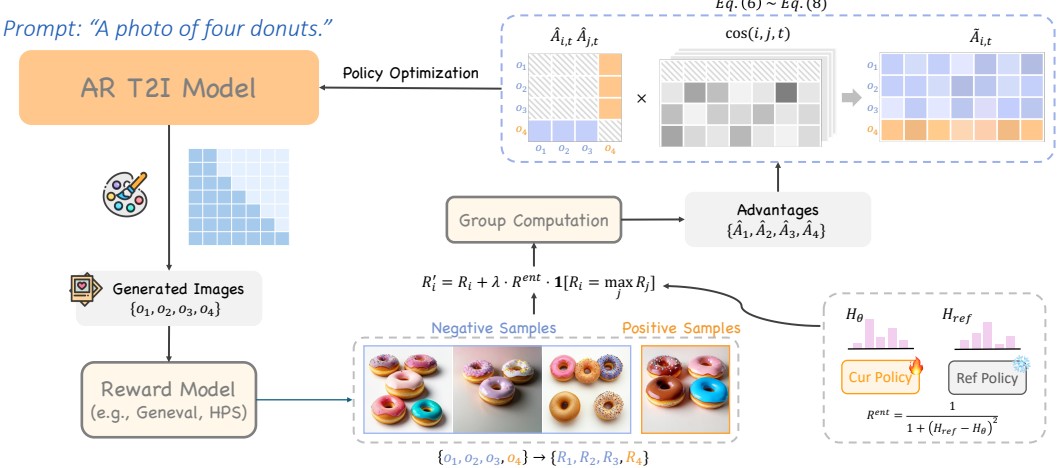

Figure 14: Overall view of our framework. At each iteration the policy generates a group of samples $o_i$ that are scored to yield sequence rewards $R_i$ and evaluated by the current and reference policies to obtain log-probs and entropies $H_\theta, H_{\text{ref}}$. The entropy reward and sequence reward combine into $R'$, which is normalized to advantages $\hat{A}$, reweighted to $\tilde{A}$ with token KL weights, and used with importance ratios $r_{i,t}(\theta)$ to form the GRPO objective $J_{\text{GRPO}}(\theta)$.

## B  DETAILED EXPERIMENTAL SETTING

### B.1  IMPLEMENTATION DETAILS

**Training pipeline** Following the setup of T2I-R1, we adopt Janus-Pro 7B as the base model. Images are generated at a resolution of $384 \times 384$. The batch size is set to 8 (i.e., 8 prompts per step), with a group size of 8 (8 images per prompt). During GRPO inference, we apply classifier-free guidance (CFG) with a scale of 5, consistent with the official Janus-Pro configuration, and the sampling temperature is fixed to 1. The training is conducted on 8 H100 GPUs using DeepSpeed ZeRO-3 and the HuggingFace Transformers library, and Adam optimizer is used with $\beta_1 = 0.9$ and $\beta_2 = 0.999$.

For experiments of LlamaGen and STAR, We find that LlamaGen and STAR require relatively large learning rates. For LlamaGen, we use the stage-1 model (the 256-resolution generator) with a batch size of 4, a group size of 4, a learning rate of 5e-6, and 1,600 training iterations. For STAR, we use the 512-resolution version with a batch size of 4 and a group size of 3 due to memory limits, and train it for 1,600 iterations with a learning rate of 1e-5.

**Reward function** We evaluate several types of reward functions:

1) Geneval reward: Following Flow-GRPO (Liu et al., 2025b), we adopt Geneval rules (Ghosh et al., 2023) for scoring. Rewards are defined according to task type: for counting, $r = 1 - |N_{gen} - N_{ref}|/N_{ref}$; for color and position constraints, rewards reflect the proportion of correctly matched objects, with full match giving 1 and partial mismatch proportionally reduced. The final reward is averaged over all clauses. Geneval reward requires a larger learning rate and KL penalty to achieve stable improvements. Therefore, we set $lr = 5 \times 10^{-6}$, $\beta = 0.03$, train for 1600 steps, and disable gradient accumulation to enable faster and more stable performance gains.

2) Combination reward: T2I-R1 (Jiang et al., 2025) adopts a combination of rewards, including human preference score (HPS), object detector (GroundingDINO), and visual question answering model (GIT), linearly combined together. We follow their setting with learning rate $1e-6$, $\beta = 0.01$, gradient accumulation=2, and total training steps=1600, to achieve more stable performance improvement.

3) OCR reward: Following Flow-GRPO (Liu et al., 2025b), given a prompt, we generate images and apply an existing OCR tool—specifically, PaddleOCR—to compute the minimum edit distance $N_e$ between the rendered text and the target text. The corresponding reward is then defined as $r = \max(1 - N_e/N_{ref}, 0)$, where $N_{ref}$ is the number of characters inside the quotation marks in the prompt. We set $lr = 1 \times 10^{-6}$, $\beta = 0.01$, training for 1,600 steps and no gradient accumulation for OCR reward.

**Data construction** 1) For the Geneval benchmark, following Flow-GRPO, we adopt the Geneval-style evaluation prompts. Training data are constructed according to Janus-Pro's accuracy on different categories, with the ratio single_object:two_object:counting:colors:position:color_attr set to 0:1:7:1:4:5. 2) For the mixed reward setting, we follow T2I-R1 and use the same prompts, consisting of 6k+ text prompts with GPT-4o-mini extracted objects and attributes from T2I-CompBench (Huang et al., 2023) and Guo et al. (2025). 3) For the OCR reward, we use the training and test sets provided by Flow-GRPO, which consist of raw image prompts containing text renderings generated by GPT-4o, includes 20K training prompts and 1K test prompts.

## C  ADDITIONAL QUANTITATIVE RESULTS

### C.1  ADDITIONAL EXPERIMENTS

**Generalization experiments with GenEval reward**. Here we provide additional generalization experiments on the GenEval reward. Specifically, we compute several image quality–related metrics on the HPS prompts and DrawBench prompts (see Table 6). The original GRPO algorithm tends to collapse during training, which harms the quality of generated images and even yields lower scores than the original Janus-Pro on certain metrics. Adding an entropy reward helps stabilize training and further improves image quality. In addition, our proposed similarity-based dynamic weighting scheme also contributes to image quality improvements.

Table 6: Generalization on Geneval benchmark for models trained with the Geneval reward on HPS and drawbench.

|  | HPS↑ | ImageReward↑ | Pickscore↑ | DeQA↑ | Aesthetic↑ |
|---|---|---|---|---|---|
| Janus-Pro-7b | 28.64 | 0.76 | 21.83 | 3.53 | 5.68 |
| T2I-R1 | 29.83 | 0.94 | 22.03 | 3.65 | 5.91 |
| Baseline | 28.49 | 0.67 | 21.86 | 3.49 | 5.55 |
| +Ent | 28.72 | 0.74 | 21.87 | 3.53 | 5.65 |
| Ours | 28.72 | 0.80 | 21.91 | 3.54 | 5.65 |

**Metrics during training with reward on (Jiang et al., 2025)**. Fig. 10 in the main text shows the evolution of GenEval metrics during training with the GenEval reward. Here, we provide the GenEval and HPS metrics over global steps using the hyperparameters from (Jiang et al., 2025) (Fig. 15). Compared to the baseline, dynamic weighting accelerates the improvement of HPS and stabilizes GenEval metrics, preventing the late-stage decline observed in the baseline.

**Generalization experiments with reward on (Jiang et al., 2025)**. In the main text, we reported results on T2I-CompBench and a subset of image-quality metrics (HPS, ImageReward, etc.). Here we additionally provide the corresponding GenEval scores to further substantiate the generalization of the proposed method; see Table 7 and Table 8. Compared with the baseline model, the dynamic similarity weighting strategy helps improve image quality.

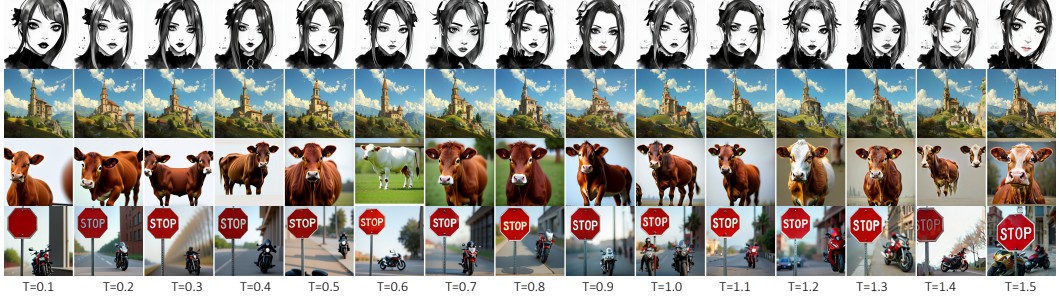

Figure 15: Evolution of HPS and GenEval metrics during training with HPS+Gdino+Git rewards. Compared to the baseline, our method achieves faster HPS improvement, more stable GenEval gains, and no noticeable late-stage decline.

Table 7: Generalization on GenEval for models trained with the HPS + GIT + Grounding DINO reward on T2I CompBench style prompts. The baseline yields almost no gain on the GenEval score of Janus Pro, whereas our method under the same setting increases GenEval from 0.78 to 0.81.

| Model | Overall↑ | Single Obj.↑ | Two Obj.↑ | Counting↑ | Colors↑ | Position↑ | Attr. Binding↑ |
|---|---|---|---|---|---|---|---|
| Janus-Pro-7B (2025) | 0.78 | 0.98 | 0.86 | 0.56 | 0.89 | 0.76 | 0.63 |
| T2I-R1 (2025) | 0.79 | 0.99 | 0.91 | 0.53 | 0.91 | 0.76 | 0.65 |
| Baseline | 0.79 | 0.98 | 0.86 | 0.57 | 0.86 | 0.83 | 0.64 |
| Ours | 0.81 | 0.98 | 0.90 | 0.62 | 0.88 | 0.83 | 0.64 |

## C.2 ADAPTATION TO MORE AR MODELS

In the main text, we report only partial T2I-Compbench results for LlamaGen and STAR. Here, we provide the full T2I-Compbench results, including those for Janus-Pro 1B, in Table 9. Compared with the original base models, our method further improves multiple aspects such as image quality and structural stability.

## D ADDITIONAL ANALYSIS

### D.1 ADDITIONAL ANALYSIS OF POLICY ENTROPY

**Relation between policy entropy & generated image** Policy entropy influences generated images in two major ways: image quality (e.g., structural clarity, content richness, naturalness) and image diversity (i.e., whether a single prompt yields varied yet prompt-aligned samples). Intuitively, adjusting the temperature $\tau$ controls the shape of the logits distribution. A smaller $\tau$ leads to more conservative sampling, reducing diversity (e.g., samples become highly similar at $\tau=0.1$ in Fig. 17) and slightly degrading image quality, as shown by the Geneval and ImageReward scores in Fig. 18 and the image visualizations in Fig. 16. In contrast, a larger $\tau$ introduces more randomness, improving diversity but often causing content instability and weaker prompt-following ability.

Figure 16: Relationship between image quality and the AR transformer's probability distribution: lower temperature yields a more concentrated distribution and more accurate generations (especially for longer prompts such as HPS prompts; see the top two rows). An overly conservative sampling policy can still degrade image quality (e.g., GenEval prompts; see the bottom two rows). Temperatures that are too high reduce performance across prompt types.

Table 8: Evaluation for models trained with the HPS + GIT + Grounding DINO reward on drawbench.

| | ImageReward↑ | Pickscore↑ | DeQA↑ | Aesthetic↑ |
|---|---|---|---|---|
| Janus-Pro-7b | 0.76 | 21.83 | 3.53 | 5.68 |
| T2I-R1 | 0.94 | 22.03 | 3.65 | 5.91 |
| Baseline | 0.93 | 22.07 | 3.65 | 5.83 |
| Ours | 0.98 | 22.09 | 3.69 | 5.89 |

Table 9: Our method shows superiority over baseline GRPO on LlamaGen and STAR, particularly on benchmarks involving spatial reasoning and counting.

| Model | Color↑ | Shape↑ | Texture↑ | 2D-Spat.↑ | 3D-Spat.↑ | Non-Spat.↑ | Numeracy↑ | Complex↑ |
|---|---|---|---|---|---|---|---|---|
| Janus-Pro 1B | 0.3505 | 0.2301 | 0.2817 | 0.1073 | 0.1916 | 0.2819 | 0.2145 | 0.2730 |
| Baseline | 0.7883 | 0.5598 | 0.7131 | 0.3495 | 0.3923 | 0.3130 | 0.5468 | 0.3844 |
| Ours | 0.7863 | 0.5629 | 0.7142 | 0.3637 | 0.3906 | 0.3129 | 0.5663 | 0.3860 |
| Origin (LlamaGen) | 0.4248 | 0.3928 | 0.5041 | 0.0813 | 0.2406 | 0.3047 | 0.3761 | 0.4406 |
| Baseline (GRPO) | 0.4721 | 0.4187 | 0.5326 | 0.0769 | 0.2488 | 0.3025 | 0.3901 | 0.4745 |
| Ours | 0.4746 | 0.4238 | 0.5554 | 0.0961 | 0.2818 | 0.3063 | 0.4109 | 0.4846 |
| Origin (STAR) | 0.5570 | 0.4438 | 0.5393 | 0.1627 | 0.3408 | 0.3073 | 0.4987 | 0.5134 |
| Baseline (GRPO) | 0.6039 | 0.4941 | 0.5849 | 0.1967 | 0.3931 | 0.3142 | 0.5249 | 0.5610 |
| Ours | 0.5957 | 0.5032 | 0.6013 | 0.2063 | 0.4002 | 0.3156 | 0.5456 | 0.5667 |

**Policy entropy during RL** From the experiments in Fig. 18, for the chosen base model, a more peaked probability distribution benefits both the aesthetic quality of generated images and their text-following ability. Therefore, under the T2I-r1 reward configuration (HPS + Grounding DINO + GIT) and under the OCR reward, entropy tends to decrease steadily (See Fig. 19), yielding a more precise distribution that improves the corresponding generation metrics. In addition, because RL is on-policy and trains the policy model using images generated by the same policy model, the generation policy gradually becomes conservative, and the entropy itself tends to decrease.

For the GenEval reward, as discussed in the main text, its rule-based and discrete scoring mechanism means that, for many prompts, reshaping the probability distribution yields little change in the GenEval reward. As a result, policy entropy fluctuates during RL, making the original model distribution more prone to distortion and limiting RL efficiency (See Fig. 19). A KL loss can mitigate this to some extent, but an overly strong KL loss can restrict model performance and encourage reward hacking. By contrast, the proposed entropy reward effectively alleviates this issue.

### D.2 ADDITIONAL ANALYSIS OF ENTROPY REGULARIZATION

**Entropy reward or entropy loss.** Alternatively, entropy may be incorporated directly into the loss—for example, as a $D_{\mathrm{KL}}$-style term similar to Eq. 3 in mainpaper. However, this strategy is difficult to tune and often induces instability during training. Specifically, following Eq. 11 in the main text, we construct an additional loss term, denoted as $D_{\mathrm{ent}}$:

$$\Delta \mathcal{H}_i = \mathcal{H}_{\mathrm{ref}}(o_i) - \mathcal{H}_\theta(o_i), \tag{13}$$

$$\mathcal{L}_{\mathrm{ent}} = \lambda \cdot \frac{1}{G} \sum_{i=1}^{G} \left( \Delta \mathcal{H}_i \right)^2. \tag{14}$$

Here $\lambda$ denotes the coefficient. We find that implementing entropy regularization as a loss term is difficult to constrain: regardless of how small $\lambda$ is set, entropy tends to increase in late-stage training (see Fig. 20), and the KL loss throughout training remains higher than when using our entropy-reward scheme. This may be because the loss term applies the entropy penalty uniformly to all samples, which can shift optimization away from the highest-reward samples and thus reduce the controllability of the RL process.

**Why only reward at top-rewarded samples.** As noted in Eq. 12, we apply the entropy reward only to the top-rewarded samples because uniformly adding it can bias optimization (e.g., samples with low GenEval but high entropy reward). In experiments where the entropy reward was applied to all samples, training became less stable: entropy and KL loss fluctuated more and GenEval performance deteriorated (see Fig. 21 and Fig. 22).

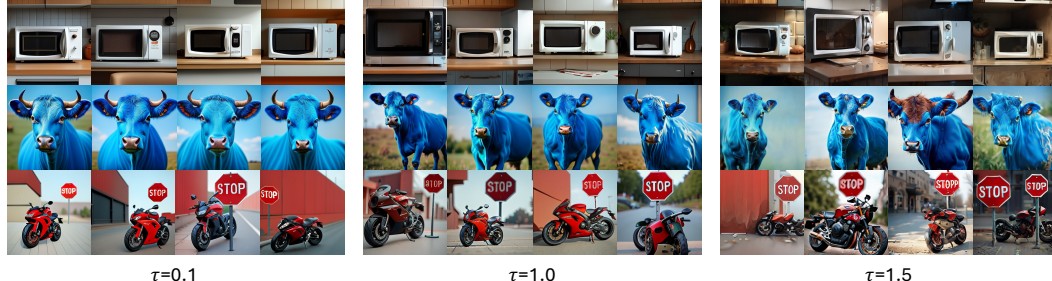

| | | |
|---|---|---|
| $\tau=0.1$ | $\tau=1.0$ | $\tau=1.5$ |

Figure 17: The relationship between diversity and logits distribution by controlling temperature $\tau$. A low temperature reduces diversity by making the distribution overly peaked, whereas a high temperature increases diversity by smoothing the distribution but may compromise prompt-following ability.

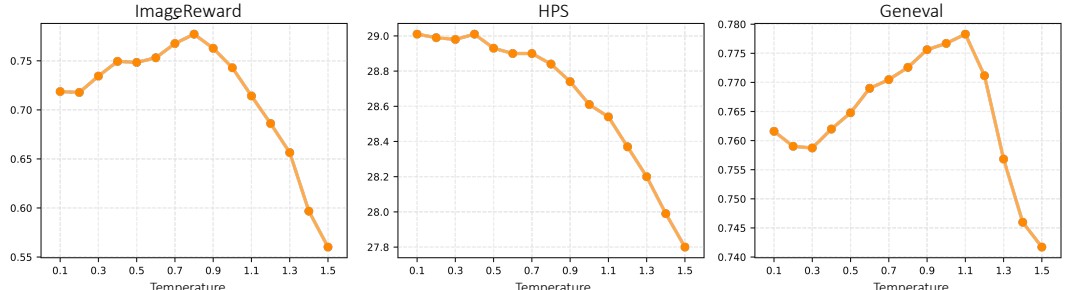

Figure 18: Relationship between evaluation metrics and the shape of the AR transformer's probability distribution: adjusting the sampling temperature controls the distribution's shape. Lower temperature makes the distribution more concentrated and the sampling policy more conservative, whereas higher temperature flattens the distribution and increases exploration.

**Replacing entropy reward with existing reward.** We further examine the effect of entropy reward by replacing it with a continuous reward model (HPS). As shown in Fig. 23, during training the GenEval score increases slightly at the beginning but then keeps decreasing, while the HPS score improves significantly compared with using entropy as the reward. By analyzing the KL loss and policy entropy, we find that the HPS reward distorts the base model distribution early in training, causing a sharp drop in entropy, which in turn makes it difficult for the GenEval score to improve in later stages.

### D.3 ADDITIONAL ANALYSIS OF IMAGE DIVERSITY

The proposed similarity-based weighting method computes the dynamic weight by measuring the average similarity between positive and negative samples generated from the same prompt rollouts. This naturally raises a concern: weighting by similarity could potentially reduce diversity. To investigate this, we further analyzed the samples produced by the policy model after RL. Interestingly, we found that in some scenarios, similarity weighting may even prevent similarity collapse during RL (see Fig. 24).

This is because the RL process in AR image generation inherently tends to stabilize the sampling policy and reduce diversity, especially in the later training stages when the original model distribution is significantly distorted. Our proposed method, however, can partly preserve the original probability distribution, enabling the model to learn from reward signals while maintaining more of the initial distribution's diversity. In contrast, the baseline tends to deviate more strongly from the original distribution in order to fit the reward model's preferences.

Nevertheless, diversity is still sensitive to training hyper-parameters such as learning rate and KL regularization strength. In long-term training, our method may also lead to diversity reduction. To mitigate this, we recommend performing RL fine-tuning of AR models within shorter training horizons to avoid large distribution shifts caused by prolonged optimization.

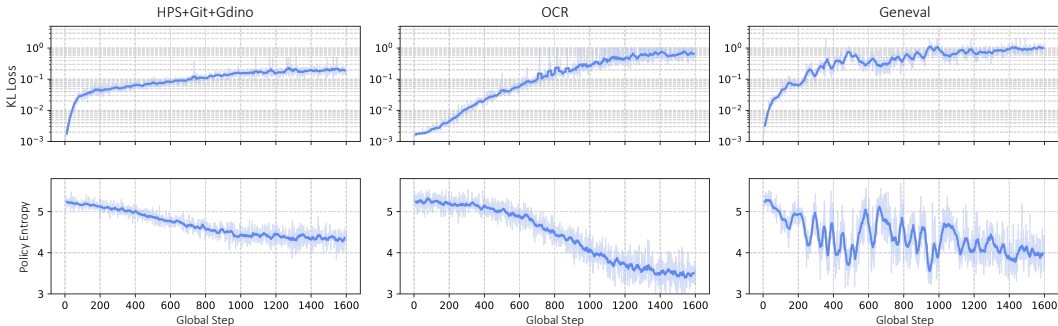

Figure 19: During training across different reward types, KL loss and policy entropy evolve with global step. For rewards with relatively continuous scoring (HPS + GIT + Grounding DINO, OCR), both curves change more smoothly; in contrast, with the GenEval reward, policy entropy and KL loss exhibit noticeable fluctuations.

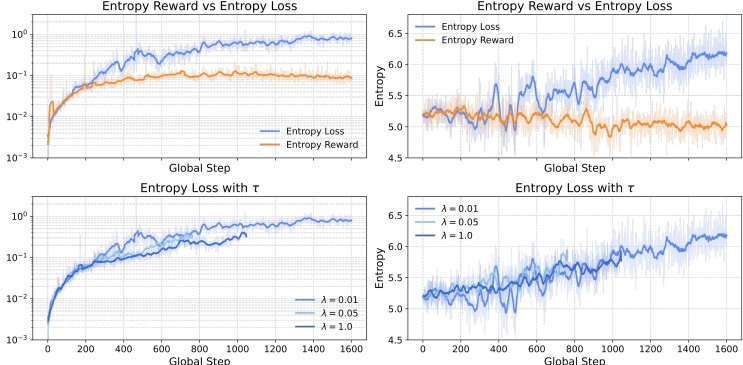

Figure 20: Entropy regularization applied as a loss term (coefficient $\lambda$) leads to late-stage entropy increase and consistently higher KL loss compared to our entropy-reward scheme.

# E LIMITATIONS & FUTURE WORKS

## E.1 LIMITATIONS

In AR visual RL tasks, instability and reward hacking remain persistent challenges. While this work alleviates them to some extent, careful tuning of hyperparameter balances is still required to avoid degraded outcomes, and achieving long-term stable training for AR models remains an open problem. Furthermore, our results highlight the importance of assigning different update magnitudes to individual image tokens in RL-based image generation. However, the current approach may reduce diversity, and exploring alternative token-selection or weighting strategies could further benefit training.

## E.2 FUTURE WORKS

For autoregressive (AR) generation with GRPO, although entropy reward and a KL coefficient can yield relatively stable training in practice, potential instability and image degradation risks remain. These issues stem from RL's inherent instability and from the fact that AR generation is highly sensitive to the shape of the underlying probability distribution. How RL can reliably improve AR visual generation therefore requires further investigation. Moreover, more general problems—such as designing better rewards to balance multiple objectives and selecting training data to balance different prompt types—also merit urgent study.

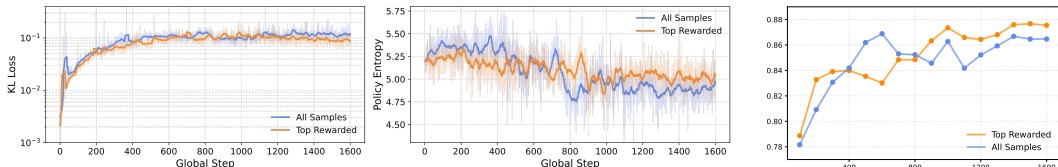

Figure 21: Compared to applying the entropy loss only to the top-rewarded samples ("Top Rewarded"), applying it to all samples ("All Samples") instead tends to cause instability during training.

Figure 22: GenEval during training with entropy reward on top vs. all samples.

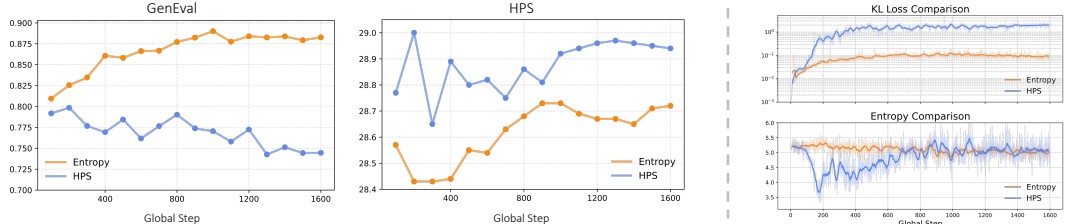

Figure 23: Replacing entropy reward ("Entropy") with HPS reward ("HPS") causes GenEval scores to drop after a brief rise, despite HPS itself improving. Early in training, policy entropy collapses and KL loss spikes, indicating distribution drift that prevents stable GenEval improvement.

## F LLM USAGE STATEMENT

We use large language model (ChatGPT) as a general-purpose assist tool. Its role and our safeguards are as follows:

- Scope of use. The LLM was used only for (i) polishing the wording of some paragraphs and (ii) drafting small visualization scripts (e.g., Python/Matplotlib plotting utilities).
- No substantive contribution. The LLM did not contribute to research ideation, problem formulation, model or experiment design, analysis, or conclusions. It is therefore not a contributor.
- Verification. All text and code suggested by the LLM were reviewed, edited, and verified by the authors. Final responsibility for the content rests with the authors.
- Data and privacy. We did not upload proprietary, sensitive, or personally identifiable data to the LLM. Only non-sensitive manuscript text and high-level coding prompts were used.

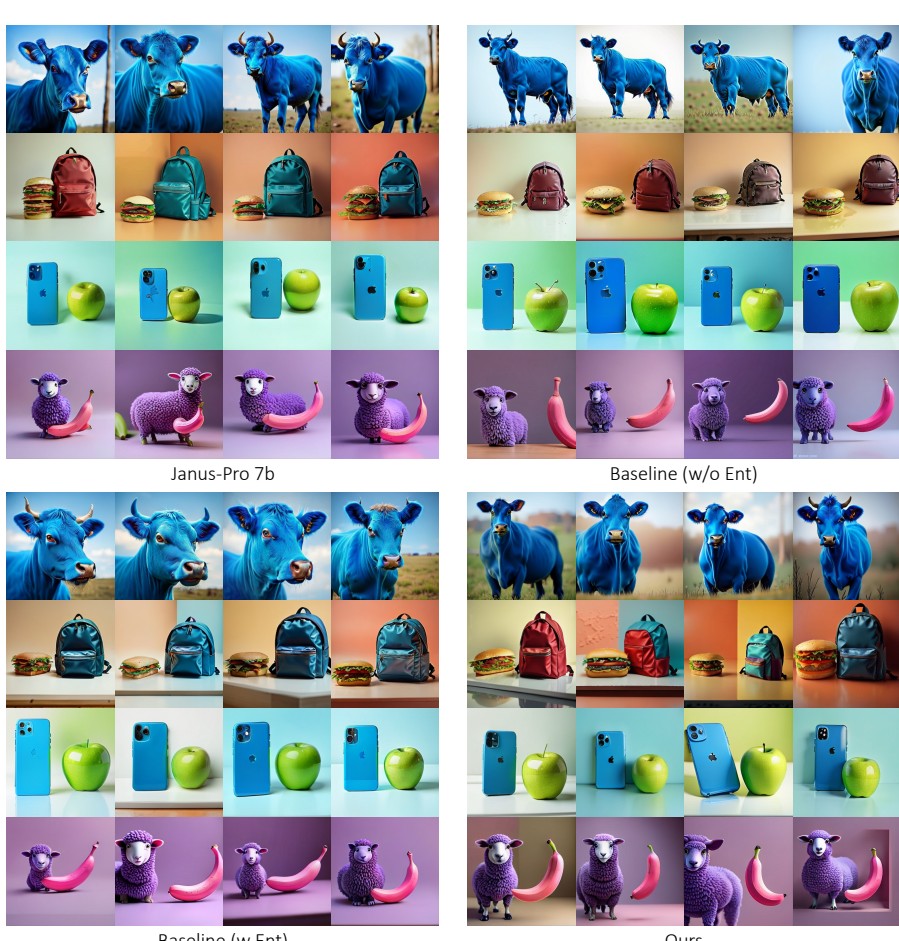

Figure 24: Visualization of generation diversity before and after RL. Given the same prompt, Janus-pro produces relatively diverse images, while GRPO ("w/o Ent") reduces diversity and frequently degrades image quality. Introducing an entropy reward recovers image quality to some extent ("w Ent"), and dynamic-weight RL yields the greatest variety ("Ours"). Notable differences include cow pose, backpack color, phone versus apple size, and sheep and banana angles.

## G    ADDITIONAL VISUAL COMPARISON

We provide additional visualizations here. Fig. 25 shows visual results of models trained with the GenEval reward on DrawBench (see Appendix Table 6 for corresponding metrics). Visualizations of models trained with HPS+Gdino+Git rewards on T2I-compbench are presented in Fig. 26 (see Table 2 in the main text and Table 7 in appendix). Finally, further visual examples for HPS and DrawBench are given in Fig. 27, with associated quantitative results reported in Table 3 (main text) and Appendix Table 8.

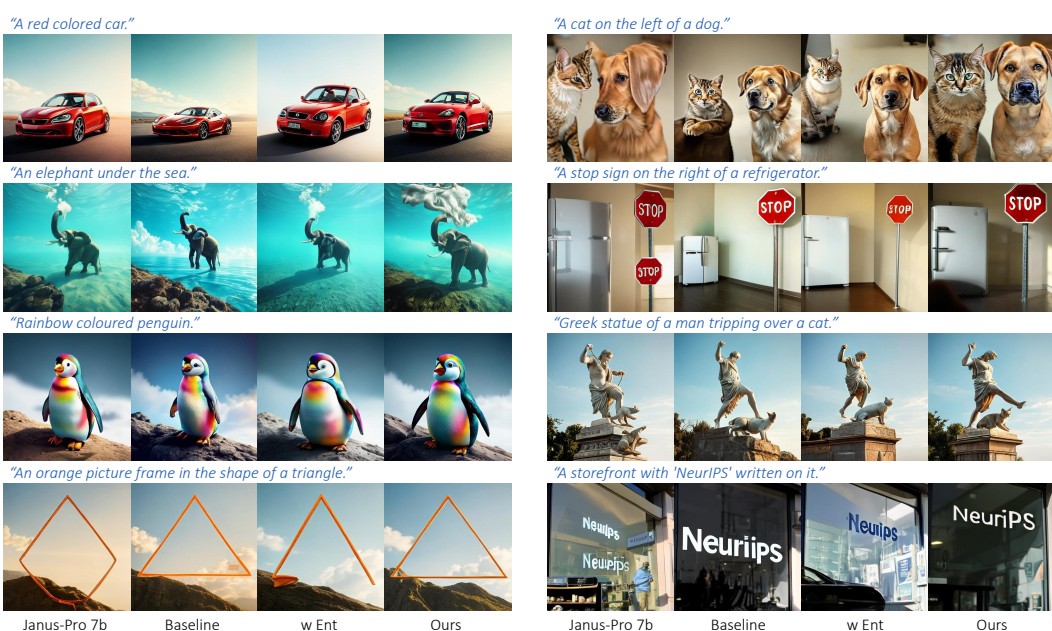

Figure 25: Visualization of models trained with the GenEval reward on DrawBench. By better reconciling the original model distribution with the reinforcement learning process, our method improves the fidelity and quality of generated images compared to the baseline.

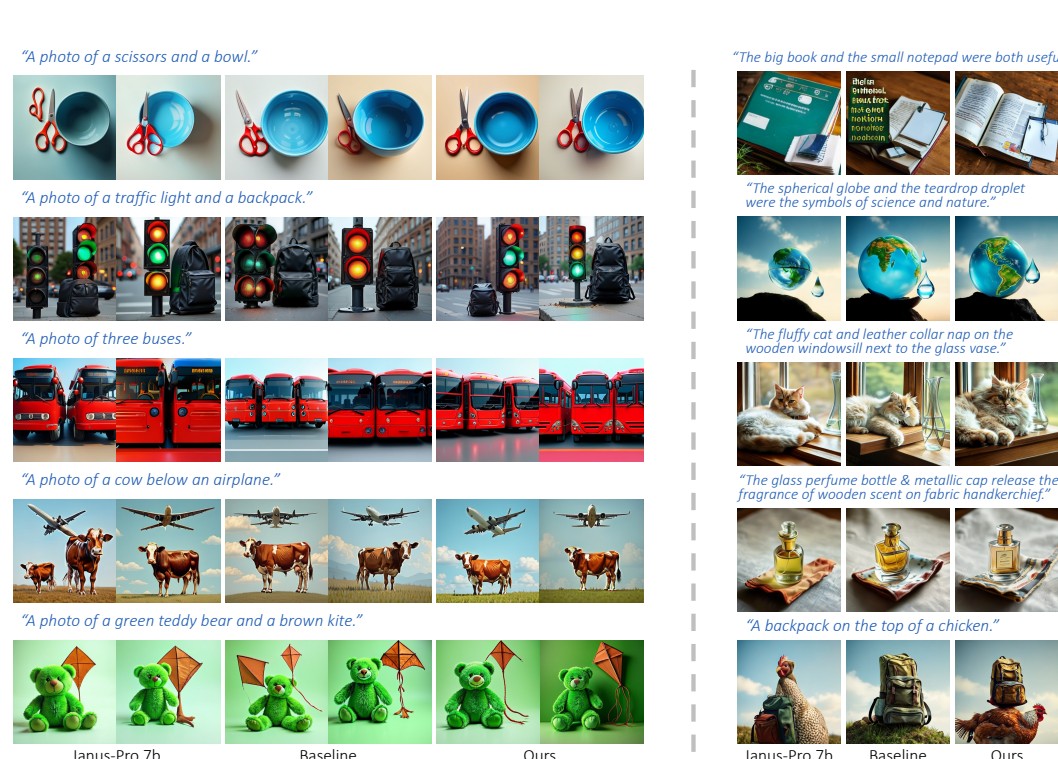

Figure 26: Visualization results of models trained with HPS+Gdino+Git rewards on T2I-compbench and GenEval.

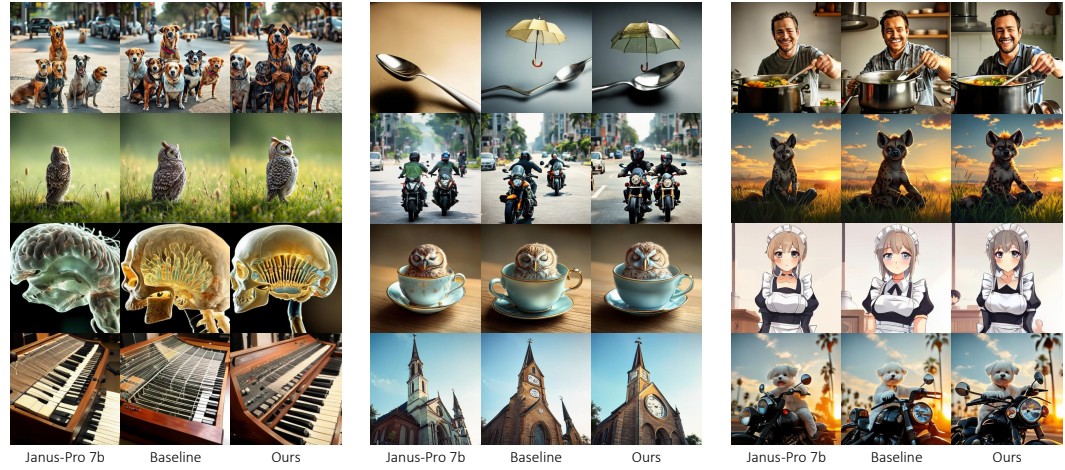

Figure 27: Visualization results of models trained with HPS+Gdino+Git rewards on drawbench and HPS prompts.

