# OpenReview forum: "STAGE: Stable and Generalizable GRPO for Autoregressive Image Generation"
_ICLR.cc/2026/Conference — Submitted to ICLR 2026_

### Official Review · Reviewer_hFsz · 2025-10-31

**Soundness:** 3
**Presentation:** 2
**Contribution:** 2
**Rating:** 4
**Confidence:** 3

**Summary:**

This paper introduces several techniques to enhance GRPO training for autoregressive (AR) image models. First, the authors observe that even when two generated images have opposite advantage values, many of their corresponding tokens remain similar. This causes conflicting gradients for those tokens, leading to unstable training. To address this issue, the authors propose to downweight the advantages of such tokens and to increase their KL regularization. Second, to mitigate the problem of low diversity, they add the entropy of token distributions as an additional reward term. Experimental results demonstrate that the proposed techniques improve both image quality and training stability.

**Strengths:**

* The observation about contradictory gradients is insightful and may have broader applications beyond this specific problem.

* The proposed methods are simple and easy to implement.

**Weaknesses:**

* The presentation of the paper—particularly in the experimental section—requires significant polishing. There are currently many unclear points; see the questions below for details.

* The evaluation is limited to a single model, which makes it difficult to assess the generality of the proposed approach.

**Questions:**

## Questions about Presentation and Clarity

* Line 236: The equation includes a clip function, but the clipping range is not specified.

* Section 3.3: The paper claims that the entropy reward is designed to stabilize training. However, based on the description and the experimental evidence, it appears to primarily encourage greater sample diversity—related to, but not equivalent to, training stability.

* Line 257: The authors state that "As shown in Fig. 2, decreasing τ leads to more deterministic samples". Whether samples are deterministic can only be evaluated with multiple generations per prompt. However, Figure 2 provides only one image per temperature. Appendix D.1 (Figure 13) referred there also shows just one image per temperature per prompt.

* Line 257 (continued): The claim that "As shown in Fig. 2, decreasing τ .. may lower image quality, while increasing τ enriches content at the cost of structural accuracy." is not clearly supported by Figure 2. I do not observe a significant difference in image quality or structural accuracy across temperatures, and the HRS scores show no clear trend.

* Figure 4: The prompt "A photo of a bench left of a" appears incomplete.

* Eq 11: Intuitively, R_i^{ent} could be any decreasing function of \delta H. Why was this particular form chosen?

* If I understand correctly, the proposed approach requires a set of prompts. Which dataset of prompts was used in training?

* Line 353: The phrase "the latter applied only for GenEval reward" is unclear. Does it mean that the entropy reward is applied only when using the GenEval reward? If so, I have two follow-up questions:
    * Line 323 states that all experiments use "GenEval rules as the reward." However, the above sentence implies other rewards are used as well. Please clarify.
    * Why is the entropy reward not applied to other rewards? Does it hurt performance or is there another reason?

* Line 361: The sentence "Additionally, the model trained with HPS+Gdino+Git mixed reward (“Baseline” and “Ours”) further demonstrates the stability improvement of structures and layouts in generated images." reads awkwardly. It does not reference specific results, and HPS+Gdino+Git is discussed in more detail only in the following paragraph.

* Line 364: The discussion here involves additional rewards, but Line 323 previously stated that all experiments use "GenEval rules as the reward." This inconsistency is confusing.

* Line 375: The sentence "For models trained with GenEval reward, we evaluate effects of different training settings on T2I-Compbench, ImageReward and provide visualizations on GenEval" repeats information already covered in the preceding "T2I-Compbench" paragraph.

* Line 404: The statement "Removing the entropy reward or dynamic reweighting negatively affects image quality and generalization; by contrast, our method better balances performance and generality." does not point to specific results, and the detailed discussion actually appears later in Section 4.3.

* Figure 9: The caption claims that all three methods show worse diversity. Without visual comparison to the base model, this conclusion is not well-supported.

* The naming conventions throughout the experiments are confusing. For example:
    * Figures 9 and 10 define "Full" as "Baseline + Entropy reward." This is unconventional—"Full" typically implies all proposed components are included, yet this version omits advantage and KL reweighting.
    * Does "Ent" in the method names (e.g., in Table 4) denote "Entropy reward"?
    * Table 4 labels "w/o Ent" as "Baseline." I thought "Baseline" is referred to the base Janus-Pro 7B model with vanilla GRPO. If "Baseline" actually means "w/o Ent," then the naming "Baseline + Entropy reward" in Figures 9 and 10 becomes self-contradictory.
    * In Table 4, what is the difference between "w/o Reweight" and "w/o KL"? Does "w/o Reweight" remove both advantage and KL reweighting, while "w/o KL" removes only KL reweighting?
    * In Table 4, does "w/o Ent & KL" mean that only advantage reweighting is applied?

## Questions about Experiments

* Line 236: The parameters a and b are fixed to 0.5. Please clarify the rationale behind these choices and/or provide a sensitivity analysis.

* The paper only evaluates the method on a single model (Janus-Pro 7B). Does the approach generalize to other models?

* How do advantage reweighting and KL reweighting individually affect performance? An ablation study would clarify their respective contributions.

## Questions about Related Work Discussion

The discussion on parallel multi-token generation (Section 2.1) omits an important line of work--Distilled Decoding--which enables generating all tokens in a single pass (see [ICLR 2025, https://arxiv.org/abs/2412.17153 ] and [NeurIPS 2025, https://arxiv.org/abs/2510.21003 ]). While the NeurIPS 2025 paper have appeared too recently to be known to the authors, the ICLR 2025 paper has been publicly available for quite some time.

## Summary

Overall, I find the experimental section difficult to follow, and I am unable to fully understand or assess the reported results at this stage. Consequently, I am assigning a negative score for now. Once the authors clarify these issues, I will carefully re-read the experimental section and am open to re-evaluating my score based on the revised version.

---

> ### Author Response · Authors · 2025-11-24
> **Rebuttal (1/4)**
>
> ***Questions about Presentation and Clarity***
> >**Q1**: Line 236: The equation includes a clip function, but the clipping range is not specified.
>
> The clip function operates in the range [0,1], while Sim(i,t) ranges from [-1,1].
> After clipping, all negative similarities collapse to 0, meaning the KL weight becomes `a`
> when `Sim(i,t) < 0`, and `a + b·clip(Sim(i,t))` otherwise. We have corrected this in the new version of the manuscript.
>
> >**Q2**: Section 3.3: The paper claims that the entropy reward is designed to stabilize training. However, based on the description and the experimental evidence, it appears to primarily encourage greater sample diversity—related to, but not equivalent to, training stability.
>
> Thank you for pointing this out. You are correct that training stability and sample diversity are not identical concepts, though they can influence each other. In AR image RL, once the original policy distribution is distorted, the model tends to overfit the newly sampled rollouts, which leads to both reduced diversity and degraded image quality. By helping preserve the base model's distribution throughout training (i.e., improving stability), the entropy reward prevents such distribution drift and allows RL to inject new knowledge without collapsing diversity.
>
> >**Q3**: The authors state that "As shown in Fig. 2, decreasing τ leads to more deterministic samples". Whether samples are deterministic can only be evaluated with multiple generations per prompt. However, Figure 2 provides only one image per temperature. Appendix D.1 (Figure 13) referred there also shows just one image per temperature per prompt.
>
> We provide additional qualitative examples in **Appendix D.1** to illustrate the relationship between temperature and sample diversity. As shown in **Fig. 17**, when the temperature is low (e.g., $\tau = 0.1$), diversity collapses — particularly for categories such as *cow* and *microwave*, where object layout and position remain almost fixed across samples. Increasing $\tau$ improves diversity, while overly large $\tau$ leads to unstable and incoherent generations.
>
> >**Q4**: Line 257 (continued): The claim that "As shown in Fig. 2, decreasing τ .. may lower image quality, while increasing τ enriches content at the cost of structural accuracy." is not clearly supported by Figure 2. I do not observe a significant difference in image quality or structural accuracy across temperatures, and the HRS scores show no clear trend.
>
> We updated the $\tau$ range shown in Fig. 2 by adding samples with $\tau = 1.2$ and $\tau = 1.5$. We observe that both settings—especially $\tau = 1.5$—produce clear structural distortions, and their entropy rewards are correspondingly lower. Although HPS assigns a higher score to the $\tau = 1.5$ sample than to the $\tau = 1.2$ sample, our entropy reward still correctly identifies that the $\tau = 1.2$ sample is better than the $\tau = 1.5$ sample.
>
> >**Q5**: Figure 4: The prompt "A photo of a bench left of a" appears incomplete.
>
> Thank you for catching this. We have corrected it in the updated version of the manuscript.
>
> >**Q6**: Eq 11: Intuitively, R_i^{ent} could be any decreasing function of \delta H. Why was this particular form chosen?
>
> You are right that, in theory, any decreasing function of $\delta H$ could be used. We chose this particular form simply because we want the penalty to grow gradually as the entropy deviates more from the reference, while giving only very small reward/penalty when the entropy difference is small. This helps prevent entropy from drifting too far away from the reference model, while still keeping the RL training itself mostly unaffected.

---

> ### Author Response · Authors · 2025-11-24
> **Rebuttal (2/4)**
>
> >**Q7**: If I understand correctly, the proposed approach requires a set of prompts. Which dataset of prompts was used in training?
>
> Thank you for the question. The prompts used for training are described in detail in the revised manuscript in **Appendix B.1**. Briefly:
>
> 1) For GenEval reward, we follow Flow-GRPO and construct prompts based on the Janus-Pro evaluation rules. Both the baseline GRPO and our method use exactly the same prompt set.
>
> 2) For HPS+GIT+Gdino reward, we follow the prompt setup used in T2I-R1, ensuring consistent evaluation and a fair comparison.
>
> 3) For OCR, we also follow Flow-GRPO and use prompts containing text-rendering instructions.
>
> >**Q8**: Line 353: The phrase "the latter applied only for GenEval reward" is unclear. Does it mean that the entropy reward is applied only when using the GenEval reward? If so, I have two follow-up questions:
> >
> > - Line 323 states that all experiments use "GenEval rules as the reward." However, the above sentence implies other rewards are used as well. Please clarify.
> > - Why is the entropy reward not applied to other rewards? Does it hurt performance or is there another reason?
>
> 1) Yes, in all experiments presented in the main paper, the entropy reward is used only with the GenEval reward.
>
> 2) We apologize for the ambiguity.
> - All main experiments (Table 1) are conducted using GenEval reward, and the first part of the Implementation Details section refers exclusively to this setting.
> - Other reward types (HPS, ImageReward, OCR) are treated as additional experiments, and are discussed later in Implementation Details and in **Appendix Section B.1**.
> We have revised the wording to make this clearer.
>
> 3) There are two reasons:
> - Firstly, entropy instability uniquely appears with GenEval reward. Based on our analysis **(Supplementary Fig. 19),** HPS, ImageReward, and OCR rewards exhibit stable entropy dynamics during RL.
> In contrast, GenEval produces frequent and sharp entropy oscillations, which was the motivation for introducing the entropy reward.
> - Secondly, applying entropy reward to other rewards can indeed reduce performance. For many rewards, especially HPS, lower policy entropy naturally correlates with better image quality, even without RL **(see Supplementary D.1)**.
> Adding an entropy-increasing regularizer can therefore interfere with the correct optimization direction and may hurt performance.
>
> >**Q9**: Line 361: The sentence "Additionally, the model trained with HPS+Gdino+Git mixed reward (“Baseline” and “Ours”) further demonstrates the stability improvement of structures and layouts in generated images." reads awkwardly. It does not reference specific results, and HPS+Gdino+Git is discussed in more detail only in the following paragraph.
>
> We apologize, but due to significant differences between the prompt and reward used in Generative reward training and T2I-compbench, we are also providing the results of the model trained with HPS+Gdino+Git mixed reward on T2i-compbench, as shown in line 361. The HPS & ImageReward metrics in the next section are based on the HPS+Gdino+Git mixed reward.
>
> >**Q10**: Line 364: The discussion here involves additional rewards, but Line 323 previously stated that all experiments use "GenEval rules as the reward." This inconsistency is confusing.
>
> We apologize for the lack of clarity. We have mentioned the specific experimental setup for other rewards in **Sec. 4.1 of the main text and section B.1 of Appendix**.
>
> >**Q11**: Line 375: The sentence "For models trained with GenEval reward, we evaluate effects of different training settings on T2I-Compbench, ImageReward and provide visualizations on GenEval" repeats information already covered in the preceding "T2I-Compbench" paragraph.
>
> Thank you, you're right. Both of these sections discuss the generalization performance of models trained on Generic reward on other benchmarks (T2I-compbench, Imagereward, etc.). We have reorganized the relevant descriptions.
>
> >**Q12**: Line 404: The statement "Removing the entropy reward or dynamic reweighting negatively affects image quality and generalization; by contrast, our method better balances performance and generality." does not point to specific results, and the detailed discussion actually appears later in Section 4.3.
>
> Thank you, we have merged the two descriptions into one.
>
> >**Q13**: Figure 9: The caption claims that all three methods show worse diversity. Without visual comparison to the base model, this conclusion is not well-supported.
>
> We have provided the generated image results of the base model before RL for reference in the latest draft.

---

> ### Author Response · Authors · 2025-11-24
> **Rebuttal (3/4)**
>
> >**Q14**: The naming conventions throughout the experiments are confusing. For example:
> >
> > - Figures 9 and 10 define "Full" as "Baseline + Entropy reward." This is unconventional—"Full" typically implies all proposed components are included, yet this version omits advantage and KL reweighting.
> > - Does "Ent" in the method names (e.g., in Table 4) denote "Entropy reward"?
> > - Table 4 labels "w/o Ent" as "Baseline." I thought "Baseline" is referred to the base Janus-Pro 7B model with vanilla GRPO. If "Baseline" actually means "w/o Ent," then the naming "Baseline + Entropy reward" in Figures 9 and 10 becomes self-contradictory.
> > - In Table 4, what is the difference between "w/o Reweight" and "w/o KL"? Does "w/o Reweight" remove both advantage and KL reweighting, while "w/o KL" removes only KL reweighting?
> > - In Table 4, does "w/o Ent & KL" mean that only advantage reweighting is applied?
>
> 1) We have revised all naming conventions in the updated draft; a summary table is provided below:
>
> | Model                | adv. Reweight | KL Reweight | Entropy reward | KL terms (std=0) |
> |----------------------|---------------|-------------|----------------|------------------|
> | **Janus-Pro 7B**     |               |             |                |                  |
> | Baseline             |               |             |                | ✔️               |
> | Ours                 | ✔️            | ✔️          | ✔️             | ✔️               |
> | w/o adv. Reweight    |               | ✔️          | ✔️             | ✔️               |
> | w/o KL Reweight      | ✔️            |             | ✔️             | ✔️               |
> | **Ent+KL (w/o Reweight)** |         |             | ✔️             | ✔️               |
> | w/o Ent              |               |             |                | ✔️               |
> | w/o KL               |               |             | ✔️             |                  |
> | w/o Ent & KL         |               |             |                |                  |
>
> 1) Yes, entropy reward is abbreviated as “Ent” throughout the paper. We have clarified this in the caption of Table 1.
>
> 2) “w/o Reweight” removes only the advantage/KL reweighting module. A detailed discussion of KL-related terms is provided in Sec. 4.3 (“Discussion of KL loss”).
>
> 3) “w/o Ent”, “w/o KL”, and “w/o Ent & KL” are evaluated under the setting without advantage/KL reweighting (i.e., starting from “w/o Reweight”). Therefore, “w/o Reweight” means removing both advantage and KL reweighting, and the variants further remove entropy reward and/or group-std KL masking.
>
> 4) “w/o Ent & KL” is closest to Baseline without the KL-masking heuristic (removing KL terms when reward std = 0), and is used to examine whether weakening KL constraint alone can improve performance.
>
>
> ***Questions about Experiments***
>
> >**Q15**: Line 236: The parameters a and b are fixed to 0.5. Please clarify the rationale behind these choices and/or provide a sensitivity analysis.
>
> We note that Sim ranges in [-1, 1]. Thus, when Sim(i,t) < 0, the KL weight reduces to `a`, and when Sim(i,t) ≥ 0, it becomes `a + b · clip(Sim(i,t))`. We set `a + b = 1` so that the maximum KL penalty matches the original formulation, while `a` controls the minimum penalty applied when similarity is negative. Due to time constraints, we are still running additional experiments and will include a more complete analysis in a future revision.
>
> >**Q16**: The paper only evaluates the method on a single model (Janus-Pro 7B). Does the approach generalize to other models?
>
> We have provided more AR models in the new version (including LlamaGen based on next-token prediction and STAR based on next-scale prediction; due to space constraints, we have listed them in a simple table here):
>
> | Model      | Setting     | **T2I-Compbench ↑** |        |        | HPS ↑ | ImgRwd ↑ | GenEval ↑ |
> |------------|-------------|----------------------|--------|--------|-------|-----------|-----------|
> |            |             | Text ↑               | 2D-Spat. ↑ | Num. ↑ |       |           |           |
> | **LlamaGen** | Pre-GRPO    | 0.5041               | 0.0813 | 0.3761 | 21.27 | -0.31     | 0.32      |
> |            | Baseline    | 0.5326               | 0.0769 | 0.3901 | 22.53 | -0.22     | 0.35      |
> |            | Ours        | 0.5554               | 0.0961 | 0.4109 | 22.74 | -0.17     | 0.39      |
> | **STAR**     | Pre-GRPO    | 0.5393               | 0.1627 | 0.4987 | 26.38 | 0.51      | 0.47      |
> |            | Baseline    | 0.5849               | 0.1967 | 0.5249 | 30.17 | 0.76      | 0.49      |
> |            | Ours        | 0.6013               | 0.2063 | 0.5456 | 30.50 | 0.89      | 0.51      |
>
> Please refer to **Table 4, Fig. 9, and Appendix Table 9 for additional results and Sec. B. 1** for implementation details.
>
> LlamaGen: Autoregressive Model Beats Diffusion: Llama for Scalable Image Generation
> STAR: Scale-wise Text-to-image generation via Auto-Regressive representations

---

> ### Author Response · Authors · 2025-11-24
> **Rebuttal (4/4)**
>
> >**Q17**: How do advantage reweighting and KL reweighting individually affect performance? An ablation study would clarify their respective contributions.
>
> We discussed the performance impact of removing advantage and KL reweighting separately. For models trained on the GenEval reward, both lead to a decrease in the GenEval metric. However, on some benchmarks (such as T2I-compbench and ImageReward), we unexpectedly found that removing advantage reweighting actually performs better. Experimental results and related analyses have been updated in the new draft, see **Sec. 4.3 "Impact of advantage & KL reweighting"**. A brief table is provided below to illustrate the situation:
>
> | Model             | GenEval ↑ | T2I-Compbench Color ↑ | Shape ↑ | Text. ↑ | 2D-Spat. ↑ | Num. ↑ | ImgRwd ↑ |
> |-------------------|-----------|------------------------|---------|---------|-------------|---------|-----------|
> | Janus-Pro 7B      | 0.78      | 0.6355                 | 0.3494  | 0.4929  | 0.1931      | 0.4423  | 0.76      |
> | Ours              | 0.88      | 0.7463                 | 0.4388  | 0.6443  | 0.3053      | 0.5278  | 0.80      |
> | w/o adv. Reweight | 0.87      | 0.7460                 | 0.4544  | 0.6507  | 0.3120      | 0.5294  | 0.82      |
> | w/o KL Reweight   | 0.86      | 0.7349                 | 0.4107  | 0.6226  | 0.2965      | 0.5058  | 0.76      |
>
> Our further analysis suggests that both approaches improve performance through similar mechanisms: advantage reweight controls the degree of updates in different regions based on similarity, while KL reweight reduces constraints on updates in dissimilar regions. Both are helpful for in-distribution measurements under sufficient training. However, because advantage reweight reduces many updates to some extent, it may cause the model distribution to more closely resemble the original distribution, thus limiting out-of-distribution performance improvements.
>
> ***Questions about Related Work Discussion***
> >**Q18**: The discussion on parallel multi-token generation (Section 2.1) omits an important line of work--Distilled Decoding--which enables generating all tokens in a single pass (see [ICLR 2025, https://arxiv.org/abs/2412.17153 ] and [NeurIPS 2025, https://arxiv.org/abs/2510.21003 ]). While the NeurIPS 2025 paper have appeared too recently to be known to the authors, the ICLR 2025 paper has been publicly available for quite some time.
>
> Thank you for pointing this out. *Distilled Decoding* is indeed an excellent and highly relevant work. We have added the appropriate citation in the updated version of the manuscript. Due to space constraints, we were unable to include a more detailed discussion, but we fully acknowledge its contribution and relevance to community.

---

> ### Author Response · Authors · 2025-11-28
>
> Thank you very much for the valuable reviews. We have put significant effort into responding to each comment with new analyses, experiments, and explanations. We hope our rebuttal resolves your concerns, and we would be grateful for any further feedback or questions from the reviewers. Thank you again for the time you have dedicated to our submission.

---

### Official Review · Reviewer_tgvN · 2025-10-31

**Soundness:** 2
**Presentation:** 3
**Contribution:** 3
**Rating:** 6
**Confidence:** 2

**Summary:**

This paper introduces STAGE, a novel GRPO framework tailored for autoregressive (AR) image generation, addressing challenges like contradictory gradients and unstable entropy dynamics. By leveraging similarity-aware advantage/KL reweighting and entropy-based rewards, the proposed method enhances generalization, structural consistency, and visual quality, outperforming baseline GRPO across multiple benchmarks.

**Strengths:**

* Innovation: Pointing out the differences between AR image generation and text, identifying naive GRPO's limitations.
* Performance: Demonstrates clear improvements in generalization, structural stability, and image quality, validated across diverse benchmarks.
* Comprehensive Methodology: Combines reweighting and entropy-based rewards to tackle training instability and reward hacking in AR image generation.

**Weaknesses:**

1. Some claims lack precision. For example, the motivation states AR models are "highly sensitive to small distribution shifts," but many studies show spatial image tokens exhibit weak sequential characteristics[1][2], requiring more rigorous explanation.

2. Differences between text and image tokens in GRPO lack sufficient evidence. Additionally, entropy-based regularization has been applied to language models[3], but the paper does not clearly distinguish its contributions from prior work.

3. The correspondence between the proposed solutions and the three identified challenges is not clearly articulated, making the problem-solution mapping somewhat ambiguous.

ref:

[1]Elucidating the design space of language models for image generation

[2]Generative Multimodal Pretraining with Discrete Diffusion Timestep Tokens

[3]Visual Autoregressive Modeling: Scalable Image Generation via Next-Scale Prediction

**Questions:**

1. The paper mentions setting the entropy regularization coefficient λ to 0.4. What is the impact of this value, and does it require fine-tuning for different tasks or models?

2. Could the authors provide more empirical analysis to clarify the differences between language and image tokens in GRPO?

3. How does the method perform with varying tokenization strategies, such as VAR[4] or DDT[2], or reward functions?

ref:

[4]Beyond the 80/20 Rule: High-Entropy Minority Tokens Drive Effective Reinforcement Learning for LLM Reasoning

---

> ### Author Response · Authors · 2025-11-23
> **Rebuttal (1/2)**
>
> >**Q1**: Some claims lack precision. For example, the motivation states AR models are "highly sensitive to small distribution shifts," but many studies show spatial image tokens exhibit weak sequential characteristics[1][2], requiring more rigorous explanation.
>
> Thank you for pointing this out. Our statement that AR models are “highly sensitive to small distribution shifts” refers specifically to next-token prediction-based AR image generators, in contrast to continuous diffusion models.
>
> AR models generate images sequentially, where each token is sampled from a probability distribution conditioned on all previously generated tokens. Small perturbations in the logits or early token predictions accumulate through the sequence, amplifying errors over hundreds of steps—a phenomenon also noted in other works [1,2]. This discrete, autoregressive nature makes RL training more fragile compared to diffusion models, which operate on continuous latent spaces and establish global structure early before iterative refinement.
>
> To demonstrate this sensitivity more rigorously, we add controlled Gumbel noise to token logits during AR generation **(Sec. 3.2 "Differences between text and image tokens" and Fig. 2(a))**. Even mild noise leads to noticeably different image structures; and stronger noise disrupts the entire generation trajectory. This behavior empirically supports our claim that AR models are sensitive to distribution shifts.
>
> >**Q2**: Differences between text and image tokens in GRPO lack sufficient evidence. Additionally, entropy-based regularization has been applied to language models[3], but the paper does not clearly distinguish its contributions from prior work.
>
> **Differences between text and image tokens in GRPO:**
>
> We highlight three key properties unique to AR image generation:
>
> 1) VQ indices are not semantically discrete: We demonstrate this by replacing ~50% of image tokens with their top-50 nearest neighbors in the VQ embedding space, and yields almost unchanged images **(see Sec. 3.2 and Fig. 2(b))**. This shows that many indices encode similar visual features, unlike text tokens which carry distinct semantics.
>
> 2) Rollouts from the same prompt are extremely similar: For models like Janus-Pro, samples generated from the same prompt often share large regions. If advantages differ, GRPO may assign opposite gradients to visually identical regions simply because their token indices differ, introducing noise.
>
> 3) AR decoding is distribution-sensitive: As shown in **Sec. 3.2**, adding mild Gumbel noise to logits greatly perturbs generated images, demonstrating that small distribution shifts amplify along long AR sequences—even if spatial tokens themselves have weak local sequentiality.
>
> **Difference from entropy regularization in LLMs**:
>
> Prior LLM work uses entropy regularization mainly to avoid collapse. In contrast, we find that in AR image generation:
>
> - different rewards drive entropy in different directions (HPS tends to prefer lower entropy, see **Appendix D.1**,
>
> - image quality is tightly linked to entropy **(Appendix D.1)**, and
>
> - replacing our entropy reward with HPS causes early entropy collapse, reward hacking, and distribution drift **(Fig. 13(a) and Sec. 4.3)**.
>
> >**Q3**: The correspondence between the proposed solutions and the three identified challenges is not clearly articulated, making the problem-solution mapping somewhat ambiguous.
>
> Thank you for the comment. We clarify the one-to-one correspondence between the three identified challenges and our proposed solutions.
>
> 1) **Challenge 1** (Discrete VQ token indices correspond to continuous visual features) and **challenge 2** (Samples from the same prompt are often highly similar): These two challenges imply that purely index-based GRPO can give opposite gradients to visually similar regions with different indices. And our advantage/KL reweighting instead uses embedding similarity to modulate updates, ensuring that visually redundant regions receive reduced or consistent gradients.
>
> 2) **Challenge 3** (AR image generation is highly sensitive to distribution shifts) represents AR decoding accumulates errors along the sequence, so distribution drift can quickly destabilize training; our proposed:
> - Adv./KL reweighting reduces conflicting gradients, preventing distribution damage.
> - Entropy reward stabilizes policy entropy under GenEval reward, preventing reward hacking and collapse.

---

> ### Author Response · Authors · 2025-11-23
> **Rebuttal (2/2)**
>
> >**Question1**: The paper mentions setting the entropy regularization coefficient λ to 0.4. What is the impact of this value, and does it require fine-tuning for different tasks or models?
>
> We set λ = 0.4 mainly due to the characteristics of the GenEval reward. GenEval typically evaluates only a few rules (e.g., color, object, spatial relations), so the smallest reward gap that distinguishes different samples is usually around 0.2.
> Since our entropy reward is mean-centered before being added to the GenEval reward, choosing λ = 0.4 ensures that the entropy term does not override or distort GenEval’s scoring while still providing effective regularization.
> Therefore, this value generally does not require retuning for different models.
>
> >**Question2**: Could the authors provide more empirical analysis to clarify the differences between language and image tokens in GRPO?
>
> Thank you for the suggestion. In the revised draft, we have added a subsection **“Differences between text and image tokens” in Sec. 3.2** to provide more empirical evidence. This section further illustrates:
>
> 1) Discrete image token indices actually represent continuous visual features — replacing image tokens with their top-k nearest neighbors in the VQ embedding space still yields nearly unchanged images.
>
> 2) AR image generation is highly sensitive to distribution shifts — adding small Gumbel noise to the logits can significantly perturb the generated image.
>
> We also provide illustrative examples in **Fig. 2** to clearly demonstrate these differences.
>
> >**Question3**: How does the method perform with varying tokenization strategies, such as VAR[4] or DDT[2], or reward functions?
>
> We appreciate the reviewer’s question regarding tokenization strategies and reward functions.
> To examine whether our findings hold beyond the Janus-Pro 7B setting, we tested the proposed method on two additional AR architectures with different decoding mechanisms: **LlamaGen** [3] (next-token autoregressive generation) and **STAR** [4] (next-scale autoregressive generation). Note that models such as Infinity and HART were not included because they do not provide a compatible discrete quantizer or public training code.
>
> Across both models, we observe consistent improvements in structural fidelity, texture quality, and aesthetic metrics. These results indicate that our method is not tailored to a particular tokenizer or reward function, but instead provides a general stabilization effect for AR image generation. A concise summary of results is provided below:
>
> | Model      | Setting     | **T2I-Compbench ↑** |        |        | HPS ↑ | ImgRwd ↑ | GenEval ↑ |
> |------------|-------------|----------------------|--------|--------|-------|-----------|-----------|
> |            |             | Text ↑               | 2D-Spat. ↑ | Num. ↑ |       |           |           |
> | **LlamaGen** | Pre-GRPO    | 0.5041               | 0.0813 | 0.3761 | 21.27 | -0.31     | 0.32      |
> |            | Baseline    | 0.5326               | 0.0769 | 0.3901 | 22.53 | -0.22     | 0.35      |
> |            | Ours        | 0.5554               | 0.0961 | 0.4109 | 22.74 | -0.17     | 0.39      |
> | **STAR**     | Pre-GRPO    | 0.5393               | 0.1627 | 0.4987 | 26.38 | 0.51      | 0.47      |
> |            | Baseline    | 0.5849               | 0.1967 | 0.5249 | 30.17 | 0.76      | 0.49      |
> |            | Ours        | 0.6013               | 0.2063 | 0.5456 | 30.50 | 0.89      | 0.51      |
>
> Please refer to **Table 4, Fig. 9, and Appendix Table 9** for additional results and **Sec. B. 1** for implementation details.
>
> >**Reference**:
> >
> > - [1] Infinity: Scaling Bitwise AutoRegressive Modeling for High-Resolution Image Synthesis
> > - [2] Anchor Token Matching: Implicit Structure Locking for Training-free AR Image Editing
> > - [3] LlamaGen: Autoregressive Model Beats Diffusion: Llama for Scalable Image Generation
> > - [4] STAR: Scale-wise Text-to-image generation via Auto-Regressive representations

---

> ### Author Response · Authors · 2025-11-28
>
> We sincerely appreciate your positive feedback — it was truly encouraging for us. We have worked hard to address all of your questions in the rebuttal, and we would be very happy to continue the discussion if you have any remaining concerns.

---

### Official Review · Reviewer_XEVk · 2025-11-01

**Soundness:** 2
**Presentation:** 3
**Contribution:** 3
**Rating:** 4
**Confidence:** 3

**Summary:**

This paper identifies several issues when adopting GRPO to image autoregressive (AR) models. Specifically, the authors firstly attribute the structural degradation of generated image to the noisy and conflicting gradient during the training. To tackle this problem, the authors propose to reweight the advantage and the KL term according to the similarity between the current token and other tokens in tha batch. Additionally, the authors identify the entropy collapse during training and add an extra term of reward which aligns the current entropy with the reference entropy. Experiments show that the proposed method surpasses the vanilla GRPO significantly across all metrics.

**Strengths:**

1. This paper identifies issues in the GRPO training process and offer targeted solutions.
2. Experimental results show that the proposed method outperforms the baseline method by a large margin.

**Weaknesses:**

1. The correlation between structural degradation and noisy gradient seems not straightforward for me. The authors should provide more direct discussions or experiments to verify it.
2. I suggest the authors directly show the gradient correlation among samples within a batch to more clearly demonstrate that the proposed method reduces gradient variance across samples generated from the same prompt, and thus leads to better performance.
3. I agree that entropy is strongly correlated with performance, where excessively high or low entropy can both lead to poor quality. However, it’s unclear to me why aligning with the reference model’s own entropy necessarily yields good results. I suggest the authors include more discussion on this point.
4. Do all experiments use the same model Janus-Pro 7B? If so, validation on other models is needed to support your claim.

**Questions:**

1. It seems that the different metrics in Table 1 are tested using models trained with different rewards. Is such in-distribution evaluation rather than task-generic evaluation a common practice in related papers? Are the baseline method evaluated in the same way?

---

> ### Author Response · Authors · 2025-11-23
> **Rebuttal (1/2)**
>
> >**Q1**: The correlation between structural degradation and noisy gradient seems not straightforward for me. The authors should provide more direct discussions or experiments to verify it.
>
> >**Q2**: I suggest the authors directly show the gradient correlation among samples within a batch to more clearly demonstrate that the proposed method reduces gradient variance across samples generated from the same prompt, and thus leads to better performance.
>
> Thank you for the valuable suggestion.
>
> We follow your advice and compute per-sample gradients within each prompt group during RL. Specifically, for each sample we obtain the gradient of the final MLP head in the AR model, normalize it, and then measure gradient variance within the group as an indicator of conflicting gradients.
>
> Due to the high memory cost of per-sample gradient extraction on a 7B model, we conduct this experiment on LlamaGen with a group size of 4. We report the variance curve over 1600 training steps and its global mean in **Fig. 13(a) and Sec. 4.3 (“Impact of advantage & KL reweighting”).** The baseline GRPO exhibits an average within-group gradient variance of around **0.006**, whereas our Advantage/KL reweighting reduces it to approximately **0.003**, confirming its effectiveness in mitigating conflicting updates. We also observe occasional spikes in our method, which we plan to investigate further.
>
> >**Q3**: I agree that entropy is strongly correlated with performance, where excessively high or low entropy can both lead to poor quality. However, it’s unclear to me why aligning with the reference model’s own entropy necessarily yields good results. I suggest the authors include more discussion on this point.
>
> Our intuition is that the entropy reward effectively acts as a sample-level KL regularizer, encouraging the policy to stay close to the reference model’s natural entropy profile. This stabilizes training: it prevents premature entropy collapse (which typically leads to reward hacking and distribution drift) while also avoiding excessively high entropy that harms image fidelity. In practice, this allows the model to benefit from longer training without degrading its generation quality.
>
> We have added more discussion in **Sec. 4.3 “Impact of entropy reward”**, highlighting why the reference model’s entropy provides a stable target and how it helps maintain a healthy balance between exploration and fidelity during RL training.
>
> >**Q4**: Do all experiments use the same model Janus-Pro 7B? If so, validation on other models is needed to support your claim.
>
> Thank you for pointing this out.
> In the updated manuscript, we have extended our validation beyond Janus-Pro 7B to ensure that our conclusions hold across different AR generative models. Specifically, we implement our method on:
>
> LlamaGen [1] — a next-token AR image generator,
>
> STAR [2] — a next-scale AR model.
>
> (We did not include Infinity because it does not use a standard discrete quantizer, and HART does not provide publicly available training code.)
>
> Results are provided in **Table 4, Fig. 9, and Appendix Table 9** in the new draft, consistently showing that our method yields models with clearer structure, sharper textures, and higher aesthetic quality across architectures.
> These experiments confirm that our approach generalizes well to multiple types of AR image generation models, not only Janus-Pro 7B.
>
> A brief summary of benchmark improvements is shown below:
>
> | Model      | Setting     | **T2I-Compbench ↑** |        |        | HPS ↑ | ImgRwd ↑ | GenEval ↑ |
> |------------|-------------|----------------------|--------|--------|-------|-----------|-----------|
> |            |             | Text ↑               | 2D-Spat. ↑ | Num. ↑ |       |           |           |
> | **LlamaGen** | Pre-GRPO    | 0.5041               | 0.0813 | 0.3761 | 21.27 | -0.31     | 0.32      |
> |            | Baseline    | 0.5326               | 0.0769 | 0.3901 | 22.53 | -0.22     | 0.35      |
> |            | Ours        | 0.5554               | 0.0961 | 0.4109 | 22.74 | -0.17     | 0.39      |
> | **STAR**     | Pre-GRPO    | 0.5393               | 0.1627 | 0.4987 | 26.38 | 0.51      | 0.47      |
> |            | Baseline    | 0.5849               | 0.1967 | 0.5249 | 30.17 | 0.76      | 0.49      |
> |            | Ours        | 0.6013               | 0.2063 | 0.5456 | 30.50 | 0.89      | 0.51      |

---

> ### Author Response · Authors · 2025-11-23
> **Rebuttal (2/2)**
>
> **Questions1**: It seems that the different metrics in Table 1 are tested using models trained with different rewards. Is such in-distribution evaluation rather than task-generic evaluation a common practice in related papers? Are the baseline method evaluated in the same way?
>
> Thank you for the question. In Table 1, Baseline, Ours, and Ours* are all trained using the GenEval reward with the same set of prompts.
> For T2I-R1 [3], the original paper uses a different reward setup (HPS + GIT + Gdino) and additionally applies semantic CoT. Since our method is implemented on the same base model and training framework as T2I-R1, a fair comparison requires aligning the training configuration.
>
> Therefore, to construct a proper baseline consistent with our setting, we remove semantic CoT from T2I-R1 and replace its reward with GenEval. This modified configuration corresponds exactly to the Baseline model reported in Table 1.
>
> Thus, all models in Table 1 trained under GenEval reward (Baseline / Ours / Ours*) are evaluated in-distribution, and the baseline is evaluated under the same protocol, ensuring a fair comparison.
>
> Additionally, using evaluation metrics that match the reward function (i.e., in-distribution evaluation) is a standard practice in T2I-RL research. For example:
>
> Flow-GRPO [4] is trained with the GenEval reward and is likewise evaluated with GenEval.
>
> T2I-R1 [3] is optimized with HPS+GIT+Gdino rewards and is evaluated using aesthetic and human-preference–related metrics aligned with those rewards.
>
> Following this established convention, Table 1 adopts evaluation settings consistent with the reward used during training. Our Baseline is evaluated under exactly the same conditions, ensuring a fair and apples-to-apples comparison. Moreover, we also evaluate models trained with the GenEval reward on additional benchmarks to assess their cross-task generalization. The results are reported in **Table 2 (Baseline† / Ours†) and Table 5**, demonstrating that our method also promotes out-of-distribution performance.
>
> >**Reference**:
> >
> > - [1] LlamaGen: Autoregressive Model Beats Diffusion: Llama for Scalable Image Generation
> > - [2] STAR: Scale-wise Text-to-image generation via Auto-Regressive representations
> > - [3] T2I-R1: Reinforcing Image Generation with Collaborative Semantic-level and Token-level CoT
> > - [4] Flow-GRPO: Training Flow Matching Models via Online RL

---

> ### Author Response · Authors · 2025-11-28
>
> We greatly appreciate the reviewers’ constructive comments and have carefully revised the paper and rebuttal to address all raised issues. We hope our responses and added experiments clarify the concerns. If there is anything further we can help clarify, please feel free to let us know—we would be glad to provide additional details. Thank you for your time and consideration.

---

### Official Review · Reviewer_EPWD · 2025-11-01

**Soundness:** 2
**Presentation:** 2
**Contribution:** 2
**Rating:** 4
**Confidence:** 2

**Summary:**

The paper proposes an approach (STAGE) that effectively stabilizes GRPO training through advantage reweighting and entropy regularization. It demonstrates improved training stability and generalization across multiple benchmarks, with clearer and more consistent image generation results.

**Strengths:**

1. It is intuitively reasonable to decrease the advantage weight at positions that have high similarity to other images.
2. The training curves reported in Figures 3, 5, and 8 demonstrate improved training stability with the proposed method.
3. The paper is well organized.

**Weaknesses:**

1. The proposed methods (advantage & KL reweighting and entropy regularization) appear to be a combination of training techniques, and the technical contribution is somewhat limited.
2.  To my understanding, the entropy reward is mainly introduced to address the issue in GenEval, where different image variations may receive identical rewards. Wouldn’t the simplest solution be to combine other existing rewards instead?
3. It also seems that the ablation study on the entropy reward was conducted only under the GenEval reward setting. Would the entropy reward also provide benefits when applied to other reward functions?

Minors:
1. In Figure 1(c), the bracket should span from $o_0$ to $o_3$.
2. In Figure 11, the signs of "fire" and "snowflake" are reversed.

**Questions:**

Please see the weaknesses.

---

> ### Author Response · Authors · 2025-11-23
>
> >**Q1**: The proposed methods (advantage & KL reweighting and entropy regularization) appear to be a combination of training techniques, and the technical contribution is somewhat limited.
>
> Thank you for the comment. We would like to clarify that although our method includes advantage/KL reweighting and entropy regularization, these components are not intended as generic training tricks, but are designed to address instabilities that are specific to autoregressive (AR) image generation:
> 1. Reweighting addresses a problem that does not exist in text RL—highly similar image regions receive opposite advantages, creating conflicting gradients. Our embedding-based reweighting is specifically designed to mitigate this and yields clear, isolated gains in ablations.
> 2. Entropy reward is not replaceable by existing rewards. Replacing it with HPS collapses policy entropy and causes reward hacking, showing that its effect is fundamentally different from simply adding another reward model.
> The components solve different instabilities and work together, enabling stable long-horizon training that existing T2I-RL methods fail to achieve.
> 3. These targeted mechanisms make the model trainable and significantly improve performance, going beyond a simple combination of techniques.
>
> >**Q2**: To my understanding, the entropy reward is mainly introduced to address the issue in GenEval, where different image variations may receive identical rewards. Wouldn’t the simplest solution be to combine other existing rewards instead?
>
> Thank you for the insightful question. Combining additional rewards (e.g., HPS) with GenEval is indeed a reasonable idea, and we appreciate the chance to clarify why this does not resolve the underlying issue.
>
> First, different rewards induce different entropy preferences. For instance, HPS tends to favor low-entropy distributions (as shown in Sec. 4.3), whereas GenEval does not. When jointly optimized, the policy entropy becomes unstable and can no longer remain aligned with the reference policy.
>
> To further examine this, we replaced our entropy reward with HPS while keeping the same selection protocol (added only to the best-GenEval samples). As reported in **Fig. 13 and Fig. 23**, policy entropy drops sharply in early training, after which the model exhibits clear reward-hacking behaviors and eventually collapses **(achieve ~0.75 finally)**. As a result, GenEval performance does not continue to improve despite using a stronger reward model.
>
> These findings show that the entropy reward is not simply another reward source—it specifically stabilizes entropy dynamics and prevents distribution drift, which existing rewards fail to achieve. More detailed analysis is provided in **Sec. 4.3 (“Impact of entropy reward”) and Appendix D.2.**
>
> >**Q3**: It also seems that the ablation study on the entropy reward was conducted only under the GenEval reward setting. Would the entropy reward also provide benefits when applied to other reward functions?
>
> Thank you for raising this point. The effect of entropy is indeed reward-dependent, because AR image models are highly sensitive to the shape of the logits distribution. As discussed in Appendix D.1, even without RL, reducing entropy can improve some benchmarks such as HPS or OCR—these metrics inherently prefer sharper, lower-entropy outputs.
>
> For this reason, applying our entropy reward under a reward like HPS may actually reduce that benchmark’s score — not because entropy regularization fails, but because the benchmark itself favors low-entropy outputs.
>
> Nonetheless, we believe entropy regularization can still be made broadly beneficial. With a more adaptive formulation (e.g., allowing entropy to gradually decrease over RL training), it may provide consistent gains across different reward types. We consider this a promising direction for future work.
>
> >**Minors**: 1. In Figure 1(c), the bracket should span from $o_0$ to $o_3$.  2. In Figure 11, the signs of "fire" and "snowflake" are reversed.
>
> Thank you for pointing this out. We have corrected these issues in the revised version of the manuscript.

---

> ### Author Response · Authors · 2025-11-28
>
> Thank you again for your thoughtful and constructive reviews. We have carefully answered all raised concerns in the rebuttal and revised draft. And we would be glad to answer if there are any remaining questions or points requiring clarification. We appreciate your time and consideration.

---

### Author Response · Authors · 2025-11-29

Dear AC：

We thank you and the reviewers for the time and care invested in reviewing our submission. Below is a summary of (1) aspects the reviewers acknowledged as strengths, (2) how we addressed concerns in our rebuttal/new draft:

***Reviewers acknowledged several strengths of our method, including:***

• Our work effectively addresses practical challenges in autoregressive (AR) image generation—specifically, issues arising from discrete tokenization and noisy gradients.

• Reviewers acknowledged the strengths of our method in improving both performance and diversity. In particular:

- 1) the similarity-weighted token strategy was considered intuitive and well-motivated.
- 2) the entropy reward was recognized as beneficial for enhancing sample diversity.

• Reviewers also noted that our empirical results—on benchmarks such as GenEval, T2I-CompBench, and others—are competitive and demonstrate meaningful improvements over baseline models.

***In addition, we have made substantial efforts to address the key concerns raised by the four reviewers:***

>**Generalization to other models (XEVk, tgvN, hFsz)**:

We applied our proposed method to additional AR models, including both next-token–based LlamaGen and next-scale–based STAR. We also added qualitative examples and quantitative metrics in the final PDF. These results consistently demonstrate that our approach brings stable improvements in structural accuracy, image aesthetics, and other quality dimensions across different AR paradigms.

>**Further analysis and clarification on gradient conflicts (XEVk)**:

We provided more direct empirical evidence by visualizing the gradient variance across samples within each batch during training. Our method substantially reduces this intra-batch gradient variance for samples generated from the same prompt, further confirming that our approach preserves the rigor of the underlying model distribution.

>**Additional analysis on the effect of the entropy reward (EPWD, XEVk)**:

We experimentally show that replacing the entropy reward with other types of rewards (e.g., HPS) does not yield similar performance gains. On the contrary, because entropy decreases rapidly in early training, using alternatives causes the GenEval metrics to fail to recover in later stages. We further explain that the effectiveness of the entropy reward originates from a sample-level KL regularization effect, which prevents early collapse of the model distribution and consequently improves final performance.

>**Clarification of experimental settings (hFsz)**:

We reorganized the entire experimental section, including clearer naming of ablation settings and more consistent descriptions. We also added a broader set of ablation studies to clarify the roles of the advantage term and KL reweighting. This greatly improves the overall consistency and readability of the paper.

We have paid significant effort during the rebuttal period, and we sincerely hope that these additions and clarifications help reshape the reviewers’ and AC’s understanding of our work. We once again thank all reviewers and the AC for their time and dedication.

---

### Meta-Review · Area_Chair_EGch · 2026-01-15

**Summary:**

This paper revisits GRPO-style RL fine-tuning for autoregressive image generators and argues that naïvely porting GRPO from text to AR images leads to training instability, quality degradation, and weak generalization. The authors identify two main issues: (i) conflicting / noisy gradients and (ii) unstable entropy dynamics. They propose STAGE, which combines similarity-aware advantage/KL reweighting and an entropy-based reward to stabilize learning dynamics. Experiments on multiple benchmarks (and, in the revision, multiple AR backbones) show improved stability, visual quality, and cross-benchmark transfer over baseline GRPO.

The method remains somewhat “engineering-heavy” and the entropy component appears reward-dependent (with limited hyperparameter sensitivity analysis). Given that some of the concerns are not fully addressed, it would be hard for reviewers to reach a consensus to accept this paper.

**Reviewer Concerns:**

Fully addressed:
- Generalization beyond a single AR model: Reviewers asked for validation beyond Janus-Pro 7B; the authors added results on additional AR paradigms
- More direct evidence for “gradient conflict / noisy gradient” claims: Reviewer XEVk requested within-batch / within-prompt gradient evidence; the rebuttal adds per-sample gradient variance analysis showing reduction under the proposed reweighting.
- Clarifications of experimental protocol and naming/readability issues: Reviewer hFsz raised many clarity inconsistencies (reward settings, prompt sets, naming of ablations, missing details like clip range); the rebuttal claims concrete fixes (explicit clip range, reorganized experiments, clarified reward usage, revised naming table, prompt source description).

Partially addressed / still outstanding:
- Perceived “training tricks” / limited technical novelty (EPWD): The rebuttal argues these components are AR-image-specific and provides additional empirical evidence, but the method still reads as a careful engineering of stabilization terms rather than a fundamentally new RL objective. This concern is mitigated but not eliminated.
- Entropy reward scope and reward-dependence (EPWD, hFsz): The authors explicitly state entropy instability is most pronounced with GenEval and that adding entropy reward to other rewards can hurt performance; this is a reasonable explanation, but it also narrows the “generality” claim of the entropy component (and suggests the approach may require reward-dependent design).
- Hyperparameter sensitivity / robustness: Questions on fixed parameters (e.g., a/b for reweighting, λ for entropy term) are only partially addressed; the authors give heuristic reasoning and note limited sensitivity analysis due to time.
- Broader coverage across tokenizers / decoding strategies / rewards: While new backbones help, some reviewer questions about varied tokenization strategies and broader reward-function sweeps remain only partially explored (the rebuttal focuses on a small set of models/rewards).

**Reviewer Scores:**

- Reviewer EPWD (original: 4 / marginally below threshold), likely increase a bit
- Reviewer XEVk (original: 4 / marginally below threshold), likely stay at 4
- Reviewer tgvN (original: 6 / marginally above threshold), likely stay at 6
- Reviewer hFsz (original: 4 / marginally below threshold), likely stay at 4

---

### Decision · Program_Chairs · 2026-01-26

Reject